# The hVps34-SGK3 pathway alleviates sustained PI3K/Akt inhibition by stimulating mTORC1 and tumour growth

Ruzica Bago[1,*], Eeva Sommer[1], Pau Castel[2], Claire Crafter[3], Fiona P Bailey[4], Natalia Shpiro[1], José Baselga[2], Darren Cross[3], Patrick A Eyers[4] & Dario R Alessi[1,**]

## Abstract

We explore mechanisms that enable cancer cells to tolerate PI3K or Akt inhibitors. Prolonged treatment of breast cancer cells with PI3K or Akt inhibitors leads to increased expression and activation of a kinase termed SGK3 that is related to Akt. Under these conditions, SGK3 is controlled by hVps34 that generates PtdIns(3)P, which binds to the PX domain of SGK3 promoting phosphorylation and activation by its upstream PDK1 activator. Furthermore, under conditions of prolonged PI3K/Akt pathway inhibition, SGK3 substitutes for Akt by phosphorylating TSC2 to activate mTORC1. We characterise 14h, a compound that inhibits both SGK3 activity and activation *in vivo*, and show that a combination of Akt and SGK inhibitors induced marked regression of BT-474 breast cancer cell-derived tumours in a xenograft model. Finally, we present the kinome-wide analysis of mRNA expression dynamics induced by PI3K/Akt inhibition. Our findings highlight the importance of the hVps34-SGK3 pathway and suggest it represents a mechanism to counteract inhibition of PI3K/Akt signalling. The data support the potential of targeting both Akt and SGK as a cancer therapeutic.

**Keywords** mTORC1; mTORC2; NanoString; PI3K and NDRG1; protein kinase inhibitors; SGK3; signal transduction inhibitors

**Subject Categories** Signal Transduction; Cancer; Molecular Biology of Disease

The EMBO Journal (2016) 35: 1902–1922

## Introduction

The majority of human tumours harbour mutations promoting inappropriate activation of the Akt kinase, and therefore, inhibitors of Akt and its upstream activators including Class I PI3K are being evaluated in many cancer clinical trials (Liu *et al*, 2009; Vanhaesebroeck *et al*, 2012; Bauer *et al*, 2015). Despite the benefit observed with these molecules, drug resistance can emerge as a result of adaptive mechanisms limiting the use of these therapies (Rodon *et al*, 2013). Therefore, understanding and reverting mechanisms of resistance to PI3K/Akt inhibitors is expected to improve the success of the treatment.

The Class I PI3K family (p110α, p110β, p110γ and p110δ) is activated in response to extracellular stimuli and phosphorylates the 3′-hydroxyl group of the inositol moiety of membrane bound phosphatidylinositol 4,5-bisphosphate (PtdIns (4,5)P$_2$) to generate PtdIns (3,4,5)P$_3$ (Vanhaesebroeck *et al*, 2001; Cantley, 2002). PtdIns(3,4,5)P$_3$ and its immediate breakdown product PtdIns(3,4)P$_2$ recruit Akt and its upstream regulator phosphoinositide-dependent kinase 1 (PDK1) to the plasma membrane via their PtdIns(3,4,5)P$_3$/PtdIns(3,4)P$_2$-binding PH domains (Mora *et al*, 2004). This induces a conformational change in Akt enabling PDK1 to phosphorylate its T-loop Thr308 residue (Alessi *et al*, 1997a,b; Stokoe *et al*, 1997; Stephens *et al*, 1998). Akt is fully activated following phosphorylation by mTORC2 (mammalian target of rapamycin complex-2) at the hydrophobic motif Ser473 residue that lies within the C-terminal non-catalytic region (Sarbassov *et al*, 2005). Ser473 phosphorylation also promotes interaction of Akt with PDK1 enhancing phosphorylation of Thr308 (Najafov *et al*, 2012). A recent study has suggested that PtdIns(3,4,5)P$_3$ stimulates Ser473 phosphorylation by binding to the PH domain of the mTORC2 complex Sin1 subunit (Liu *et al*, 2015).

Although the focus has been on the role that Akt isoforms play in mediating proliferation responses that are controlled by Class I PI3Ks, increasing evidence is accumulating that isoforms of serum and glucocorticoid regulated kinases (SGK) which share ~50% identity within their catalytic domains to Akt, also control proliferation and survival responses in cancer cells (Vasudevan *et al*, 2009; Bruhn *et al*, 2010, 2013; Pearce *et al*, 2010; Bago *et al*, 2014; Gasser *et al*, 2014). Although SGK1 and SGK2 isoforms lack a PH domain,

1  MRC Protein Phosphorylation and Ubiquitylation Unit, College of Life Sciences, University of Dundee, Dundee, UK
2  Human Oncology and Pathogenesis Program, Memorial Sloan Kettering Cancer Center, New York, NY, USA
3  Oncology iMED, AstraZeneca, CRUK Cambridge Institute, Cambridge, UK
4  Department of Biochemistry, Institute of Integrative Biology, University of Liverpool, Liverpool, UK
   *Corresponding author. Tel: +44 1382385602; E-mail: r.bago@dundee.ac.uk
   **Corresponding author. Tel: +44 1382385602; E-mail: d.r.alessi@dundee.ac.uk

they are still activated by Class I PI3Ks (Kobayashi & Cohen, 1999; Kobayashi *et al*, 1999) through their ability to induce hydrophobic motif phosphorylation via mTORC2, which in turn triggers phosphorylation of the T-loop residue by PDK1 (Biondi *et al*, 2001; Collins *et al*, 2003, 2005). SGK and Akt kinases possess similar substrate specificities with both enzymes preferring to phosphorylate Ser/Thr residues lying within Arg-Xaa-Arg-Xaa-Xaa-Ser/Thr (where Xaa is any amino acid) motifs (Alessi *et al*, 1996; Murray *et al*, 2005). Consistent with this, several Akt substrates including FOXO transcription factors (Brunet *et al*, 2001), GSK3 (Kobayashi & Cohen, 1999; Dai *et al*, 2002) and NDRG1 (Sommer *et al*, 2013) are similarly phosphorylated by SGK1. It is therefore likely that in tumours displaying activated PI3K signalling, both Akt1/2 and SGK1/SGK2 would be elevated and stimulate proliferation by phosphorylating an overlapping subset of substrates. The sensitivity of a group of breast cancer cell lines possessing mutations that activate the PI3K pathway to Akt inhibitors has been correlated with the expression of SGK1 in cell lines including ZR-75-1, CAMA-1 and T47D (utilised in this study). Interestingly, these lines all contain low endogenous SGK1 levels and are intrinsically more sensitive to Akt inhibitors than cells that express higher levels of SGK1 (Sommer *et al*, 2013).

In contrast to other SGK isoforms, SGK3 possesses an N-terminal PtdIns(3)P-binding PX domain (Virbasius *et al*, 2001; Bago *et al*, 2014) and is the only known kinase to possess a PtdIns(3)P interaction domain (Pearce *et al*, 2010). SGK3 associates with endosome membranes, where the Class III PI3K family member termed hVps34 phosphorylates PtdIns to generate a large fraction of the cellular pool of PtdIns(3)P (Gillooly *et al*, 2000; Backer, 2008; Bago *et al*, 2014). Mutations within the SGK3 PX domain that ablate PtdIns(3)P binding or treatment of cells with a hVps34 inhibitor to reduce PtdIns(3)P levels blocked SGK3 endosomal localisation and also suppressed SGK3 activity, by lowering phosphorylation of T-loop and hydrophobic motifs (Bago *et al*, 2014).

In the present study, we demonstrate that prolonged treatment of several breast cancer cell lines (ZR-75-1, CAMA-1, T47D and BT-474c) harbouring mutations that activate the Akt signalling pathway with inhibitors of Class I PI3K or Akt leads to marked upregulation of SGK3 mRNA and subsequent activation of the SGK3 protein kinase. Employing structurally diverse-specific hVps34 inhibitors (VPS34-IN1 and SAR405), we establish that SGK3 activation induced by prolonged treatment with Class I PI3K or Akt inhibitors is controlled by hVps34. Mechanistic studies reveal that PtdIns(3)P binding to the PX domain of SGK3 promotes phosphorylation and activation by its upstream PDK1 activator. Moreover, we show that following prolonged inhibition of the PI3K/Akt pathway, SGK3 could substitute for Akt and promote activation of mTORC1 and hence S6K1 by phosphorylating TSC2 at the same sites as Akt. We discover that a previously reported SGK1 inhibitor termed 14h

(Halland *et al*, 2015) also potently inhibits SGK3 ($IC_{50}$ of 4 nM) with 2.5-fold higher potency than SGK1 and in addition blocks SGK3 activation by PDK1 and mTORC2 in cells. Moreover, we show that a combination of Akt (MK-2206) and SGK (14h) inhibitors induce marked regression of tumour volume in a nude mouse xenograft model derived from BT-474 breast cancer cells. Finally, we present a kinome-wide analysis of mRNA expression dynamics induced in response to PI3K/Akt pathway inhibition. Our findings highlight the importance of the hVps34-SGK3 pathway, which is likely to represent a major mechanism that can be used by cells to counteract inhibition of the PI3K/Akt signalling network. Our results clearly emphasise the therapeutic potential of targeting both the Akt and SGK kinases for the treatment of cancer.

# Results

## Prolonged treatment with Akt and Class I PI3K inhibitors leads to upregulation of SGK3

Treatment of breast cancer cell lines ZR-75-1 and CAMA-1 that have low levels of SGK1 (Sommer *et al*, 2013) with structurally diverse Akt inhibitors (MK-2206 (Hirai *et al*, 2010) and AZD5363 (Davies *et al*, 2012)) for 1 h inhibited phosphorylation of PRAS40 (Thr246, Akt-specific substrate) as well as NDRG1 (Thr346, Akt and SGK substrate) (Fig 1A). MK-2206 is an allosteric inhibitor that suppresses Thr308 and Ser473 phosphorylation (Hirai *et al*, 2010), whereas AZD5363 is a catalytic inhibitor that although ablating Akt activity in cells leads to increased phosphorylation of Thr308 and Ser473 by modulating feedback loops (Davies *et al*, 2012). Strikingly however, prolonged Akt inhibitor treatment over 1–10 days led to a time-dependent recovery of NDRG1 phosphorylation, under conditions where the Akt substrate PRAS40 remained dephosphorylated (Fig 1A). Immunoblotting revealed that prolonged Akt inhibitor treatment enhanced expression of SGK3 protein over this period under conditions where SGK1 remained undetectable (Fig 1A). Quantitative mRNA analysis by RT–PCR revealed that Akt inhibitors induced 3- to 6-fold increase in SGK3 mRNA levels after 2–10 days, whereas SGK1 or SGK2 mRNA levels were unaltered (Fig 1B). Interestingly, knockdown of SGK3 protein expression employing 3 distinct shRNA probes blocked prolonged Akt inhibitor treatment from enhancing NDRG1 phosphorylation in both ZR-75-1 and CAMA-1 cells (Fig 1C).

Similarly, treatment of ZR-75-1, CAMA-1 and T47D cultured in serum with structurally diverse Class I PI3K inhibitors GDC0941 (Folkes *et al*, 2008) and BKM120 (Bendell *et al*, 2012) for 1 h also markedly suppressed NDRG1 and PRAS40 phosphorylation (Fig 2A). Analogous to what was observed with prolonged Akt

**Figure 1. Prolonged treatment with Akt inhibitors leads to upregulation of SGK3.**

A, B ZR-75-1 and CAMA-1 cell lines were treated with 1 μM MK-2206 or 1 μM AZD5363 for the indicated time periods. Cell lysates were subjected to (A) immunoblot analysis with the indicated antibodies or (B) mRNA isolation followed by cDNA preparation. Real-time PCR was performed on cDNA samples using specific primers against SGK1, SGK2 and SGK3 isoforms. Relative mRNA levels were calculated using $2^{(-\Delta\Delta)} C_t$ method using DMSO-treated samples as a calibrator. Results are presented as relative mRNA level means ± SD for triplicates.

C ZR-75-1 and CAMA-1 cells were treated with 1 μM MK-2206 or 1 μM AZD5363 for one hour (1 h) or 10 days. After 7-day treatment, SGK3 was knocked down by using three different shRNA probes, named SGK3 A, B and C and compared to a control shRNA probe, named sh scramble. Cells were maintained in the presence or absence of the indicated inhibitor during this period. At day 10, cells were lysed and subjected to immunoblot analysis with the indicated antibodies.

Source data are available online for this figure.

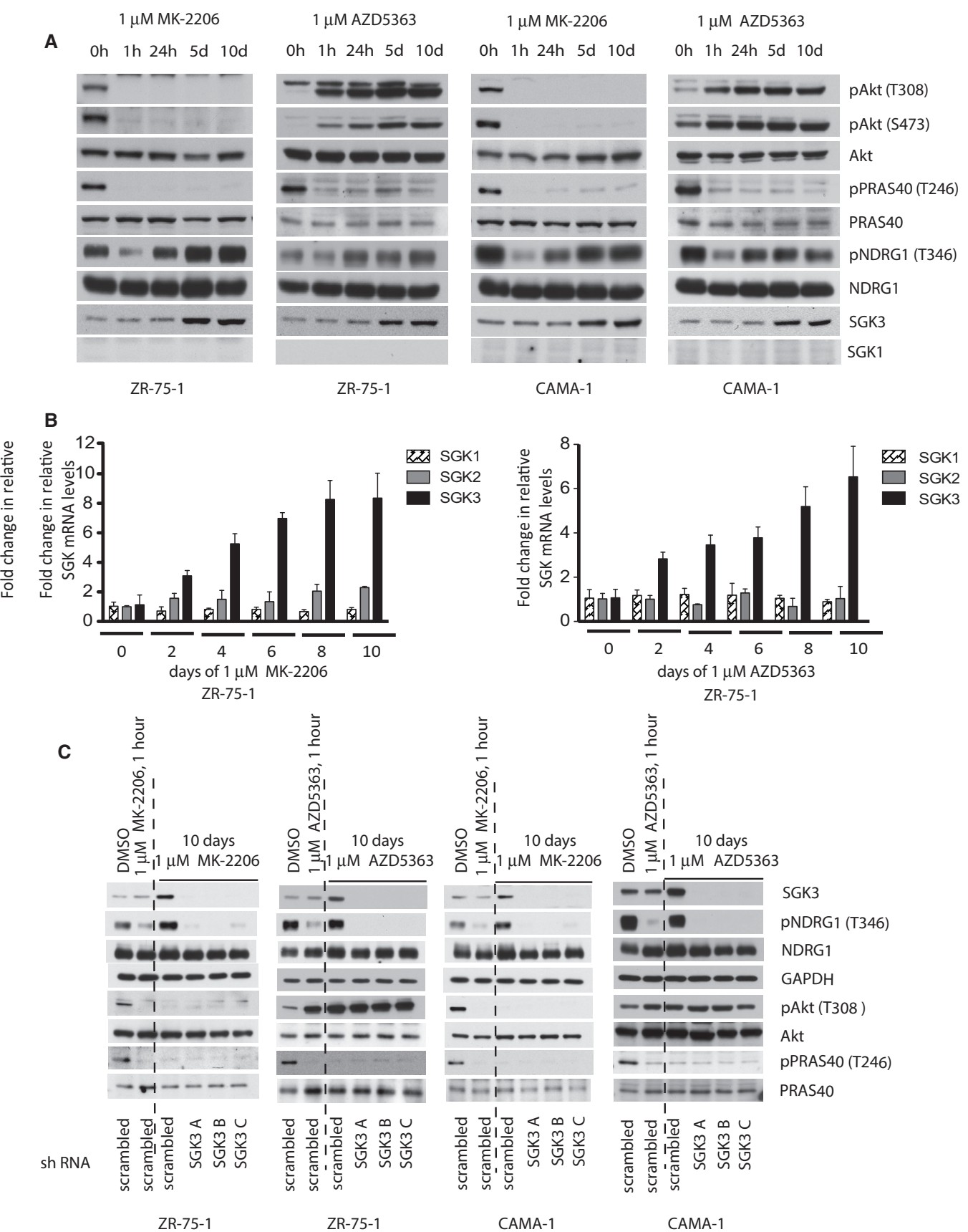

**Figure 1.**

**A**

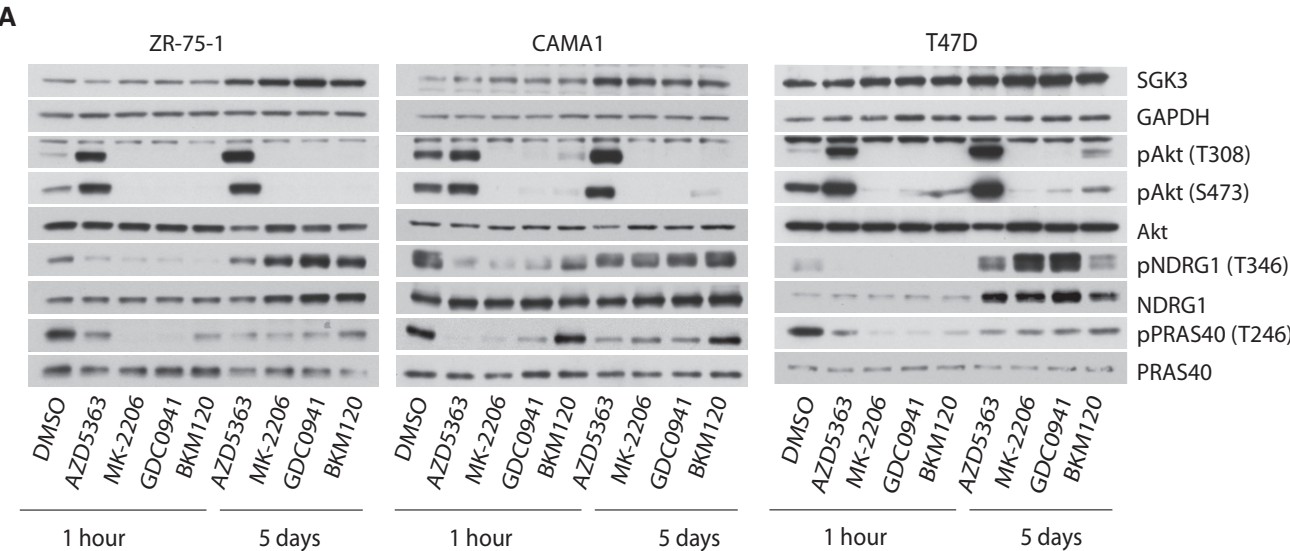

**B**

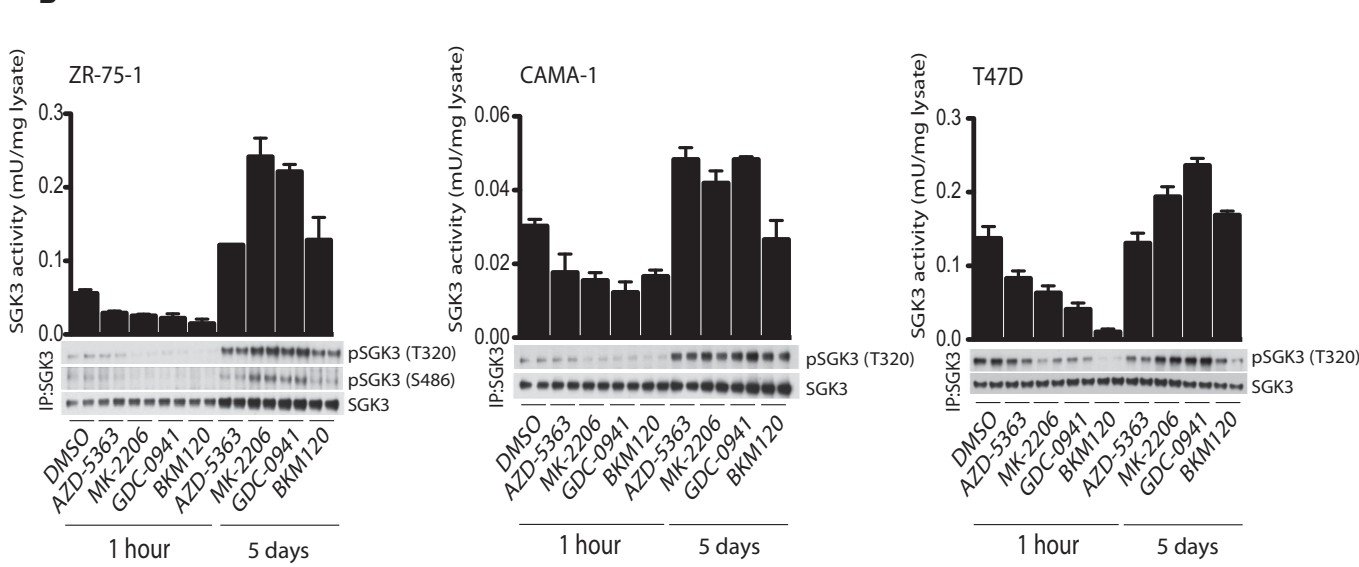

**Figure 2. Prolonged treatment with Class I PI3K inhibitors leads to upregulation of SGK3.**

A  The indicated cell lines were treated with either 1 μM MK-2206, 1 μM AZD5363, 1 μM GDC0941 or 1 μM BKM120 for the indicated times. Cell lysates were subjected to immunoblot analysis with the indicated antibodies.

B  The indicated cells were treated as in (A) and SGK3 was immunoprecipitated from the lysates using an anti-SGK3 antibody. The immunoprecipitates (IP) were subjected to *in vitro* kinase assay by measuring phosphorylation of the Crosstide substrate peptide in the presence of 0.1 mM [γ-$^{32}$P]ATP in a 30 min 30°C reaction (upper panel) followed by immunoblot analysis with the indicated antibodies (lower panel). Kinase reactions are presented as means ± SD for triplicate reaction.

Source data are available online for this figure.

inhibitor treatment (Fig 2A), prolonged treatment with Class I PI3K inhibitors induced upregulation of SGK3 mRNA and protein levels that was accompanied by an increase in NDRG1 phosphorylation (Figs 2A and EV1). The induction of SGK3 protein and mRNA levels was more pronounced in ZR-75-1 or CAMA-1 cells than T47D cells, but nevertheless in all 3 cell lines, prolonged treatment with Class I PI3K or Akt inhibitors induced a similar significant increase in NDRG1 phosphorylation.

**Prolonged treatment with Class I PI3K and Akt inhibitors leads to activation of SGK3**

To study the effect that Class I PI3K (GDC0941 and BKM120) and Akt inhibitors (MK-2206 and AZD5363) had on SGK3 kinase activity, we immunoprecipitated endogenous SGK3 and assessed protein kinase activity by measuring phosphorylation of the Crosstide substrate peptide (Bago *et al*, 2014). In all cell lines tested (ZR-75-1,

CAMA1 and T47D), short-term treatment (1 h) with Akt and Class I PI3K inhibitors induced transient reduction in SGK3 activity without affecting SGK3 protein levels. The underlying mechanism behind this is not clear. In both ZR-75-1 and CAMA-1 cells, prolonged treatment (5 days) with Class I PI3K or Akt inhibitors induced a robust 2- to 4-fold increase in SGK3 protein kinase activity (Fig 2B). More moderate increases of SGK3 phosphorylation and activity was observed in T47D cells, consistent with the lower induction of SGK3 (Fig 2A). Immunoblot analysis of SGK3 immunoprecipitates confirmed that prolonged treatment with Class I PI3K or Akt inhibitors increased SGK3 protein as well as T-loop and hydrophobic motif phosphorylation (Fig 2B).

## Class III PI3K, hVps34, controls SGK3 activity

To explore the role that hVps34 might play in regulating endogenous activity of SGK3 induced by prolonged treatment with Akt inhibitors, we treated ZR-75-1 cells for 5 days with MK-2206 and then incubated cells for 1 h with increasing doses of highly selective and structurally diverse hVps34 inhibitors, namely VPS34-IN1 (Bago et al, 2014) or SAR405 (Ronan et al, 2014). In ZR-75-1 cells pretreated with Akt inhibitor (MK-2206) for 5 days, both inhibitors induced a dose-dependent inhibition of NDRG1 phosphorylation with VPS34-IN1 reducing NDRG1 phosphorylation to basal levels at 1 μM (Fig 3A) and SAR405 at 0.3 μM (Fig EV2A), consistent with the higher potency of SAR405 inhibitor towards hVps34 (Bago et al, 2014; Ronan et al, 2014). In agreement with VPS34-IN1 or SAR405 having no inhibitory activity towards Class I PI3K, neither compound suppressed Akt phosphorylation nor PRAS40 phosphorylation (Figs 3B and EV2B). Elevated SGK3 protein kinase activity as well as T-loop and hydrophobic motif phosphorylation induced by prolonged treatment with Class I PI3K or Akt inhibitors was also suppressed by treatment with 1 μM VPS34-IN1 (Fig 3C) or 0.3 μM SAR405 (Fig EV2C). Treatment of ZR-75-1 cells cultured in serum for 1 h with 1 μM VPS34-IN1 (Fig 3C) or 0.3 μM SAR405 (Fig EV2C) also reduced the basal SGK3 activity and phosphorylation detected in these cells to below control levels.

## mTORC2 regulates activation of SGK3 downstream of hVps34

The identity of the hydrophobic motif kinase that phosphorylates SGK3 downstream of hVps34 has not been established. Since mTORC2 regulates activation of SGK1 (Garcia-Martinez & Alessi, 2008), we wished to explore whether mTORC2 also mediates SGK3 hydrophobic motif phosphorylation under conditions of prolonged treatment with Class I PI3K or Akt inhibitors. To achieve this, we generated ZR-75-1 cells in which the Rictor subunit of mTORC2 (Sarbassov et al, 2005) was knock down by ~90% employing an shRNA approach (Fig 3D). Consistent with efficient knockdown of Rictor and suppression of mTORC2 activity, Akt Ser473 phosphorylation was markedly reduced in the Rictor shRNA knockdown cells, whereas phosphorylation of S6 protein, which is controlled in a Rictor-independent manner by mTORC1, was unaffected (Fig 3D). Knockdown of Rictor also virtually ablated kinase activity as well as the hydrophobic motif phosphorylation of SGK3 induced by prolonged treatment with Class I or Akt kinase inhibitors (Fig 3D). Treatment of ZR-75-1 cells with the mTOR catalytic inhibitor AZD8055 that inhibits both mTORC1 and mTORC2 suppressed SGK3 hydrophobic motif and T-loop phosphorylation as well as SGK3 kinase activity (Fig 3E). However, mTORC1 specific inhibitor rapamycin that does not inhibit mTORC2 at this concentration had no effect on SGK3 phosphorylation or activity (Fig 3E).

## PtdIns(3)P promotes phosphorylation and activation of SGK3 by PDK1

To investigate whether PtdIns(3)P that is produced by hVps34 could promote phosphorylation and activation of SGK3, we utilised recombinant SGK3[S486E], in which the hydrophobic motif was mutated to Glu to mimic mTORC2 phosphorylation and promote phosphorylation by PDK1 (Biondi et al, 2001). SGK3 [S486E] was purified from HEK293 cells that had been treated with hVps34 (VPS34-IN1, 5 μM) and PDK1 inhibitor (GSK2334470, 5 μM) (Najafov et al, 2011) for 1 h, to ensure that SGK3 was in its dephosphorylated and inactive form. SGK3 [S486E] purified in this manner was incubated with lipid vesicles comprising phosphatidylcholine (PC) and phosphatidylserine (PS) containing either increasing concentrations of PtdIns(3)P or PtdIns. Kinase reactions were initiated by the addition of PDK1 and MgATP. Activation of SGK3[S486E] was assessed by monitoring phosphorylation of SGK3 at Thr320 (PDK1 site) and NDRG1 at Thr346 (SGK1 site). These studies revealed that activation of SGK3[S486E] beyond background levels was only observed in the presence of PS/PS vesicles containing PtdIns(3)P (Fig 4A). Even

**Figure 3. SGK3 activity induced by inhibition of PI3K/Akt is regulated by hVps34 and mTORC2.**

A  ZR-75-1 cells were treated with 1 μM MK-2206 for 5 days and then, 1 h prior to cell lysis, cells were further treated with increasing doses of VPS34-IN1. Cell lysates were subjected to immunoblot analysis with the indicated antibodies.

B  ZR-75-1 cells cultured in serum in the absence of Akt inhibitor were treated for 1 h with the indicated concentrations of VPS34-IN1. The cell lysates were analysed by immunoblot using the indicated antibodies.

C  ZR-75-1 cells were treated for 5 days with either 1 μM MK-2206, 1 μM AZD5363, 1 μM GDC0941 or 1 μM BKM120. One hour prior to lysis, the cells were incubated in the presence or absence of 1 μM VPS34-IN1. SGK3 was immunoprecipitated from lysates and subjected to in vitro kinase assay by measuring phosphorylation of the Crosstide substrate peptide in the presence of 0.1 mM [γ-$^{32}$P]ATP in a 30 min 30°C reaction (upper panel). Kinase reactions are presented as means ± SD for triplicate reaction. Immunoprecipitates (IP) and lysates were analysed by immunoblot with the indicated antibodies. One-hour (1-h) treatment with the PDK1 inhibitor GSK2334470 (Najafov et al, 2011) (1 μM) was used as a control for SGK3 inhibition.

D  ZR-75-1 cells were stably transfected with either a control shRNA vector (scrambled) or a shRNA vector that targets Rictor expression (shRictor). The cells were grown in the presence or absence of 1 μM MK-2206 or 1 μM GDC0941 for 5 days. SGK3 was immunoprecipitated from the lysates and subjected to in vitro kinase assay as in (C). Kinase reactions are presented as means ± SD for triplicate reaction. Immunoprecipitates (IP) and lysates (lower panel) were also subjected to immunoblot analysis with the indicated antibodies.

E  ZR-75-1 cells were cultured in the absence or presence of 1 μM MK-2206 for 5 days. Cells were then treated in the absence or presence of 0.1 μM AZD8055 or 0.1 μM rapamycin for 1 h. SGK3 was immunoprecipitated and subjected to in vitro kinase assay as in (C). Kinase reactions are presented as means ± SD for triplicate reaction. The immunoprecipitates (IP) and lysates were analysed with the indicated antibodies.

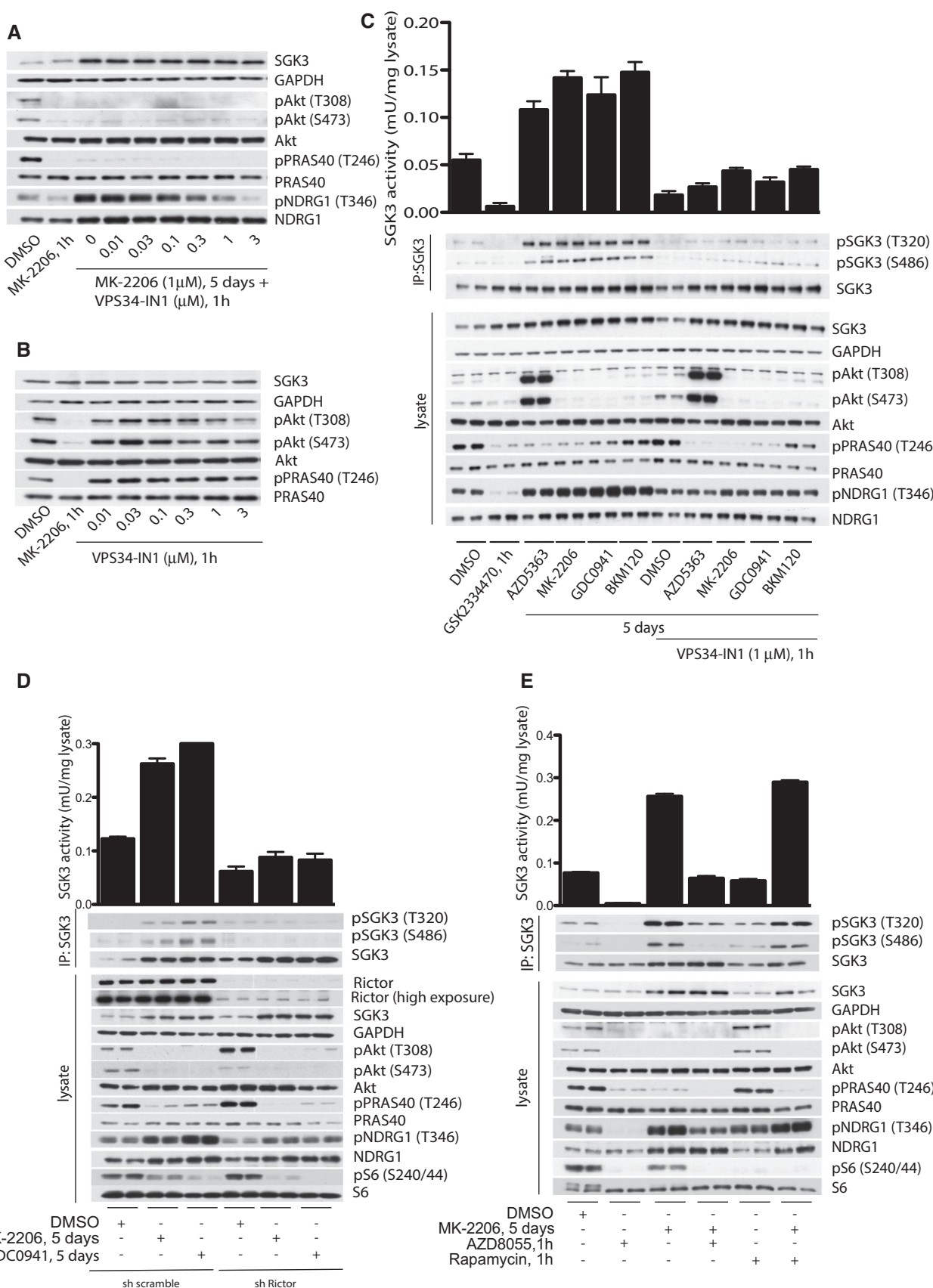

**Figure 3.**

at the highest concentration of PtdIns tested (10 μM), no significant activation of SGK3[S486E] beyond control levels was observed. We found that SGK3[S486E] isolated from HEK293 cells was contaminated with endogenous PDK1 (high exposure immunoblot, see Fig 4), which is consistent with the ability of SGK3 [S486E] to bind PDK1 with high affinity (Biondi *et al*, 2001). Therefore, even in the absence of added recombinant PDK1, incubation of SGK3[S486E] with PC/PS vesicles containing PtdIns(3)P and MgATP leads to partial activation, under conditions where no detectable activation is observed when PtdIns(3)P is replaced with PtdIns (Fig 4B). Addition of 50 ng recombinant PDK1 led to further activation of SGK3[S486E] in the presence of PtdIns(3)P

but not PtdIns (Fig 4B). Mutation of PX domain residue Arg90, required for interaction of SGK3 with PtdIns(3)P (Bago *et al*, 2014), suppressed activation of SGK3[S486E] by PDK1 in the presence of PtdIns(3)P (Fig 4B).

### 14h is a potent inhibitor of SGK3

Recently, Sanofi published a new series of SGK1 inhibitors (Halland *et al*, 2015). As the potency of these compounds towards the SGK3 isoform was not reported, we synthesised and evaluated the selectivity of the four most potent SGK1 inhibitors (14g, 14h, 14i and 14n) described in this study (Fig 5A and Appendix Fig S1A). We

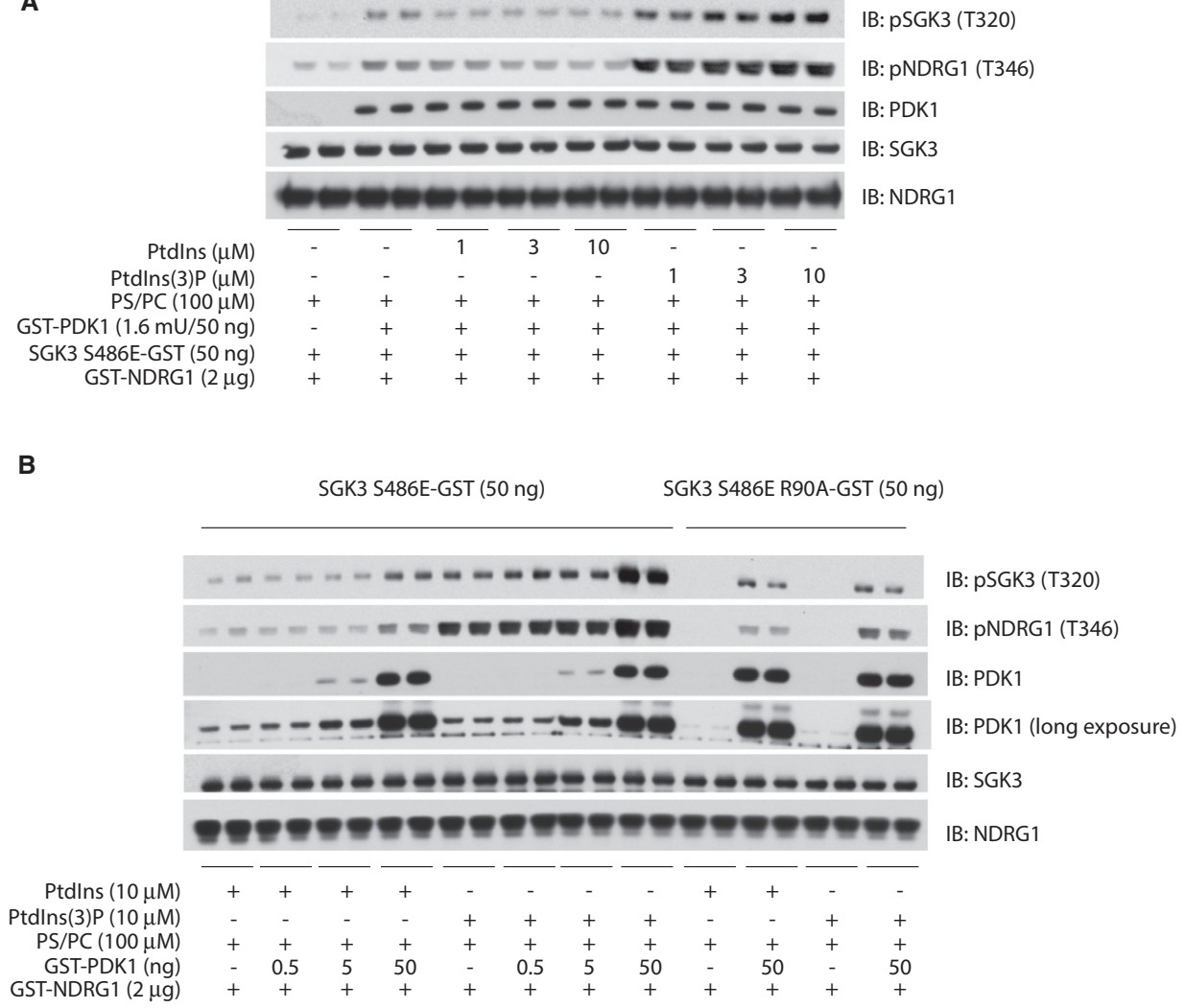

**Figure 4. PtdIns(3)P binding to PX domain of SGK3 promotes phosphorylation and activation by PDK1.**

A, B  SGK3 [S486E]-GST and SGK3 [R90A S486]-GST were purified from HEK293 cells transiently overexpressing these enzymes. One hour prior to lysis, cells were treated with 5 μM VPS34-IN1 and 5 μM GSK2334470 to ensure that the SGK3 was in its inactive dephosphorylated form. SGK3[S486E]-GST (A) or the non-PtdIns(3)P-binding mutant SGK3[R90A, S486E]-GST (B) was incubated with lipid vesicles comprising phosphatidylcholine (PC) and phosphatidylserine (PS) containing the indicted concentrations of PtdIns or PtdIns(3)P in the presence or absence of added recombinant PDK1 (50 ng) and kinase reactions were initiated by addition of MgATP. After 30 min at 30°C, PDK1 was inhibited by addition of the 1 μM GSK2334470 PDK1 inhibitor and the reaction mixture was supplemented with 2 μg GST-NDRG1 SGK3 substrate. After another 30 min at 30°C, the reaction was terminated by addition of SDS sample buffer. The reaction mixtures were subjected to immunoblot analysis with the indicated antibodies.

confirmed that these compounds inhibited SGK1 with an $IC_{50}$ of 10–70 nM and found that these compounds also targeted SGK3 with an $IC_{50}$ of 4–80 nM (Fig 5B and Appendix Fig S1B). The 14h inhibitor was the most potent against SGK3, displaying an $IC_{50}$ of 4 nM, which is 2.5-fold more potent than it inhibits SGK1 (10 nM) (Fig 5B). The selectivity of these inhibitors was assessed using the Dundee panel of 140 kinases (Fig 5C for 14h and Appendix Fig S2 for 14g, 14i and 14n and Appendix Tables S1–S4). All four SGK inhibitors displayed similar selectivity with the major off-target kinases being S6K1, MLK1, MLK3 and TIE2 (Fig 5C for 14h and Appendix Fig S2 for 14g, 14i and 14n and Appendix Tables S1–S4). We observed that the SGK inhibitors (14g, 14h, 14i and 14n) also suppressed S6K1 with $IC_{50}$ of 55–180 nM and MLK isoforms with $IC_{50}$ of 94–600 nM (Fig 5B and Appendix Fig S1B). Importantly, none of the SGK inhibitors tested significantly inhibited Akt1, suggesting that these compounds would be useful in discriminating between Akt- and SGK-mediated responses *in vivo* (Fig 5B and Appendix Fig S1B).

### 14h suppresses NDRG1 phosphorylation in cells

For cellular experiments, we employed 14h, as it was the most potent SGK3 inhibitor (Fig 5B). Treatment of ZR-75-1 cells cultured in serum with increasing doses of 14h for 1 h resulted in a dose-dependent decrease in NDRG1 phosphorylation. NDRG1 phosphorylation was maximally suppressed at 1–3 μM 14h, under conditions where Akt-specific substrate PRAS40 was not dephosphorylated. Consistent with the ~20-fold lower potency of 14h towards S6K1 compared to SGK3 (Fig 5B), 1–3 μM 14h failed to significantly inhibit Rictor (Thr1135, S6K1 specific site) and S6 protein (Ser240/244, S6K1 site) phosphorylation (Fig 5D). However, at 10 μM 14h, we noticed a moderate reduction in S6K1 and S6 protein phosphorylation (Fig 5D), suggesting that 14h should not be deployed at concentrations of higher than 3 μM in cellular studies.

### 14h suppresses T-loop and hydrophobic motif phosphorylation of SGK3

Allosteric Akt inhibitors such as MK-2206, in addition to suppressing kinase activity, also inhibit phosphorylation of the T-loop and hydrophobic motifs by trapping Akt in a conformation that cannot be phosphorylated by PDK1 and mTORC2 in cells (Green *et al*, 2008; Hirai *et al*, 2010). To determine whether 14h was capable of suppressing SGK3 activation in an analogous manner, we treated

ZR-75-1 cells with increasing doses of 14h for 1 h and analysed after immunoprecipitation SGK3 the T-loop and hydrophobic motif phosphorylation as well as kinase activity. This revealed that 14h suppressed in a dose-dependent manner SGK3 hydrophobic as well as T-loop phosphorylation resulting in reduced SGK3 activity (Fig 5E). At 0.1 μM 14h, SGK3 activity and phosphorylation were reduced by ~80% and undetectable by 1 μM (Fig 5E). In biochemical studies, 14h also suppressed the ability of PDK1 to phosphorylate SGK3[S486E] at Thr320 in the presence of PtdIns(3)P, with a near maximal inhibition observed at 0.1 μM 14h (Fig 5F).

### SGK3 activates mTORC1 independently of Akt by phosphorylating TSC2

We next treated ZR-75-1 cells with the MK-2206 Akt inhibitor for between 1 h and 5 days and at intervals measured SGK3 and S6K1 catalytic activity (Fig 6). After 1-h treatment with MK-2206 (under conditions where SGK3 activity remains low), ~10-fold reduction of S6K1 activity was observed, which was accompanied by dephosphorylation of Rictor (Thr1135) and S6 protein (Ser240/244) as well as 4EBP1 (Ser65), another key substrate of mTORC1 (Fig 6). This result is consistent with previous work showing that a major substrate of Akt phosphorylation in cancer cells is the tuberous sclerosis complex protein TSC2 (Manning *et al*, 2002; Menon *et al*, 2014), which subsequently inhibits GTPase activating protein activity of TSC2, leading to the activation of the Rheb GTPase and hence mTORC1 (Manning *et al*, 2002; Menon *et al*, 2014). Consistent with this, 1-h treatment with MK-2206 induced dephosphorylation of TSC2 (Ser939, Thr1462) at the sites phosphorylated by Akt. However, after 1–2 days of MK-2206 treatment, as SGK3 is specifically becoming upregulated, we observed a commensurate increase of S6K1 activity that correlates with increased phosphorylation of TSC2 at the Akt sites as well as Rictor, S6 protein and 4EBP1 phosphorylation, under conditions which Akt remains inactivated (Fig 6). After 4–5 days MK-2206 treatment, S6K1 activity as well as the phosphorylation of TSC2, Rictor, S6 protein and 4EBP1 had recovered to the similar level that was observed in non-Akt inhibitor-treated cells (Fig 6).

As Akt and SGK isoforms have very similar substrate specificity (Murray *et al*, 2005), our data indicate that SGK3 might functionally substitute for Akt in phosphorylating TSC2 in cells lacking Akt activity. To investigate this further, we treated ZR-75-1 cells for 5 days with either Class I PI3K (GDC0941) or Akt (MK-2206) inhibitors and investigated the effect that the 14h inhibitor had on TSC2, S6K1,

**Figure 5.    14h selectively suppresses both the activity and activation of SGK3 by PDK1 and mTORC2.**

A    Chemical structure of the Sanofi-14h SGK inhibitor.

B    $IC_{50}$ values of Sanofi-14h SGK inhibitor on the indicated recombinant kinases.

C    Protein kinase profiling undertaken against the Dundee panel of 140 protein kinases in the presence of 1 μM Sanofi-14h at the International Centre for Protein Kinase Profiling. The result for each kinase is presented as a mean kinase activity of the reaction taken in triplicate relative to a control reaction where the inhibitors were omitted. Abbreviations and assay conditions are described at http://www.kinase-screen.mrc.ac.uk.

D    ZR-75-1 cells were treated for 1 h with the indicated concentrations of 14h. The cell lysates were analysed by immunoblot analysis using the indicated antibodies.

E    ZR-75-1 cells were treated for 1 h with the indicated concentrations of 14h. SGK3 was immunoprecipitated from cell lysates and subjected to *in vitro* kinase assay by measuring phosphorylation of the Crosstide substrate peptide in the presence of 0.1 mM [γ-$^{32}$P]ATP in a 30 min 30°C reaction (upper panel). Kinase reactions are presented as means ± SD for triplicate reaction. Immunoprecipitates (IP) were also analysed by immunoblot with the indicated antibodies.

F    The effect of the indicated concentration of 14h on the ability of SGK3[S486E]-GST to be activated by PDK1 in the presence of PtdIns(3)P was assessed as described in Fig 4.

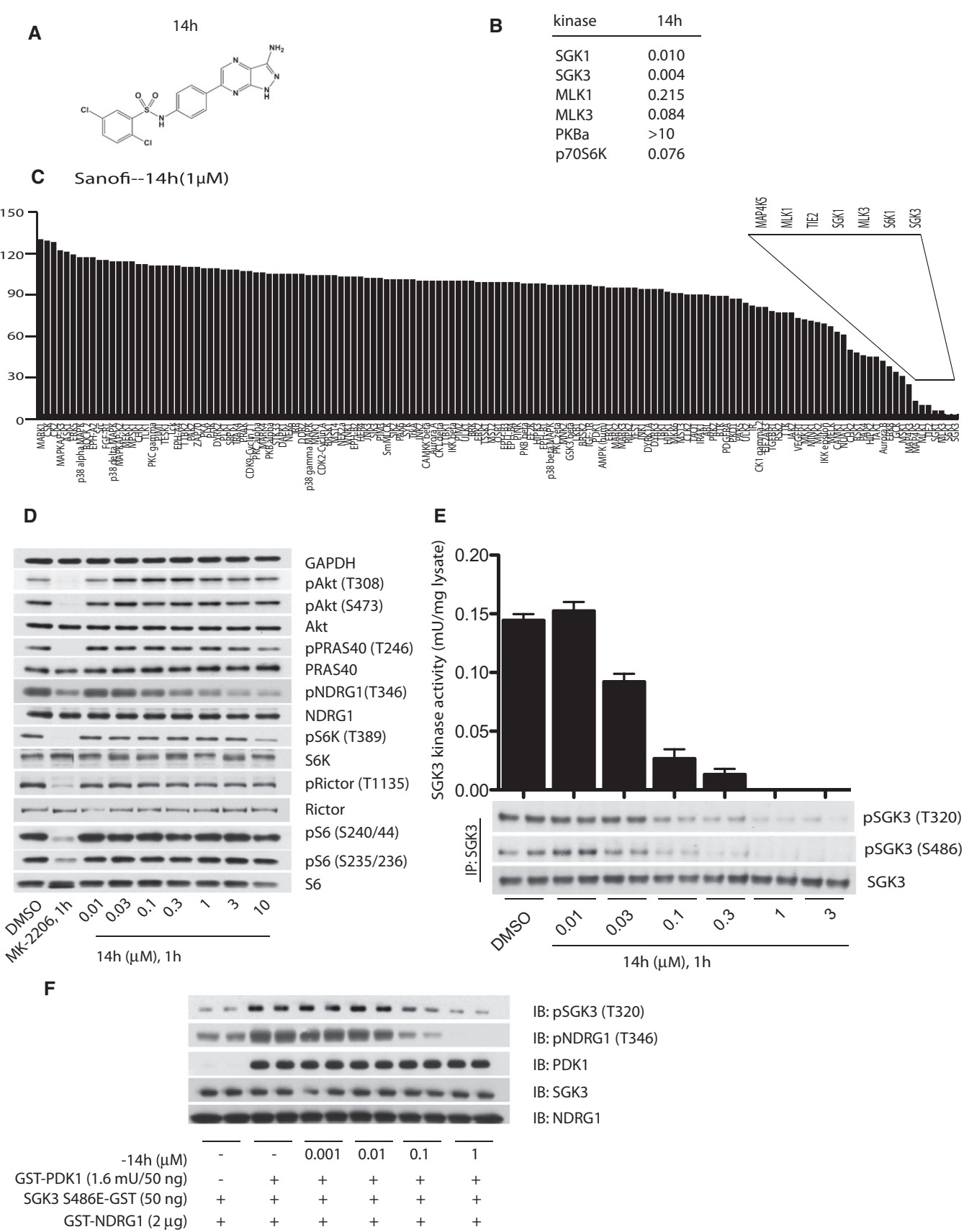

**Figure 5.**

Rictor, S6 protein and 4EBP1 phosphorylation (Fig 7). Consistent with SGK3 mediating phosphorylation of TSC2 and leading to the activation of mTORC1 under these conditions, we observed that treatment with 3 μM 14h for 1 h suppressed phosphorylation of TSC2. We also observed suppression of activity and phosphorylation of S6K1 and its downstream targets, Rictor and S6 protein (Fig 7A and B). However, phosphorylation of 4EBP1, other mTORC1 substrate, was not markedly reduced upon SGK3 inhibition (Fig 7A). In contrast, 14h did not inhibit TSC2, S6K1, Rictor, S6 protein or 4EBP1 phosphorylation or S6K1 activity in ZR-75-1 cells cultured in serum in the absence of prolonged treatment with PI3K or Akt inhibitors (Fig 7A and B). As expected, under these conditions, Class I PI3K or Akt inhibitors suppressed TSC2, S6K1, Rictor, S6 protein or 4EBP1 protein phosphorylation as well as S6K1 activity (Fig 7A) and addition of SGK inhibitor (14h) failed to ablate dephosphorylation of these substrates (Fig EV3A). To further evaluate impact of SGK3 inhibition, we performed shRNA-mediated knockdown of SGK3 expression in ZR-75-1 cells treated in the presence or absence of the Akt (MK-2206) inhibitor for 1 h or 5 days (Fig 7C). This revealed that knockdown of SGK3 markedly reduced phosphorylation of NDRG1 as well as TSC2, leading to an inhibition of mTORC1, reflected by the inhibition of S6K1, Rictor, S6 protein and 4EBP1 phosphorylation when compared to scrambled shRNA control samples (Fig 7C).

We also tested whether the addition of hVps34 inhibitor (VPS34-N1) would have the same suppressive effect on phosphorylation of TSC2 and mTORC1 substrates as SGK3 inhibition (Fig EV3B). We found that addition of 1 μM VPS34-IN1 for 1 h after prolonged treatment with Akt (MK-2206) or Class I PI3K (GDC0941) inhibitors had a similar effect as SGK3 inhibition (Fig EV3B). We observed reduced phosphorylation of TSC2, S6K1, Rictor, S6 protein and 4EBP1 (Fig EV3B). Treatment of ZR-75-1 cells for 1 h with 1 μM of VPS34-IN1 alone under conditions which SGK3 is not activated did not have an effect on TSC2 or mTORC1 substrate phosphorylation.

To explore whether any other potential SGK3 phospho-substrates could be detected, we treated ZR-75-1 cells for 1 h or 5 days with Akt inhibitor (MK-2206) or Class I PI3K (GDC0941) in the presence or absence of SGK3 inhibitor (14h) for 1 h prior to lysis. Lysates were immunoblotted with anti-p-Akt motif substrate antibody (RxRxxpS/pT) (Zhang *et al*, 2002). The results revealed that at least 6 Akt substrates detected by the p-Akt motif antibody were re-phosphorylated on Akt phosphorylation consensus motif after a 5-day exposure to Akt (MK-2206) or Class I PI3K (GDC0941) inhibitors when SGK3 is upregulated, in a manner that was blocked by the 14h SGK3 inhibitor (Fig EV3C). We also observed 3 Akt substrates that were not re-phosphorylated on Akt phosphorylation consensus motif after a 5-day exposure to MK-2206/GDC0941 when SGK3 is upregulated, suggesting these are Akt selective (like PRAS40). No evidence of any SGK3 selective substrates recognised by the p-Akt motif antibody was observed (Fig EV3C).

### Combined inhibition of both Akt and SGK is required to regress BT-474 xenograft tumours

In order to establish the therapeutic potential of combined AKT and SGK3 inhibition in a breast cancer model, we tested cell proliferation *in vitro* and established xenografts in nude mice with BT-474 cells, which are sensitive to Akt inhibitors when cultured *in vitro*

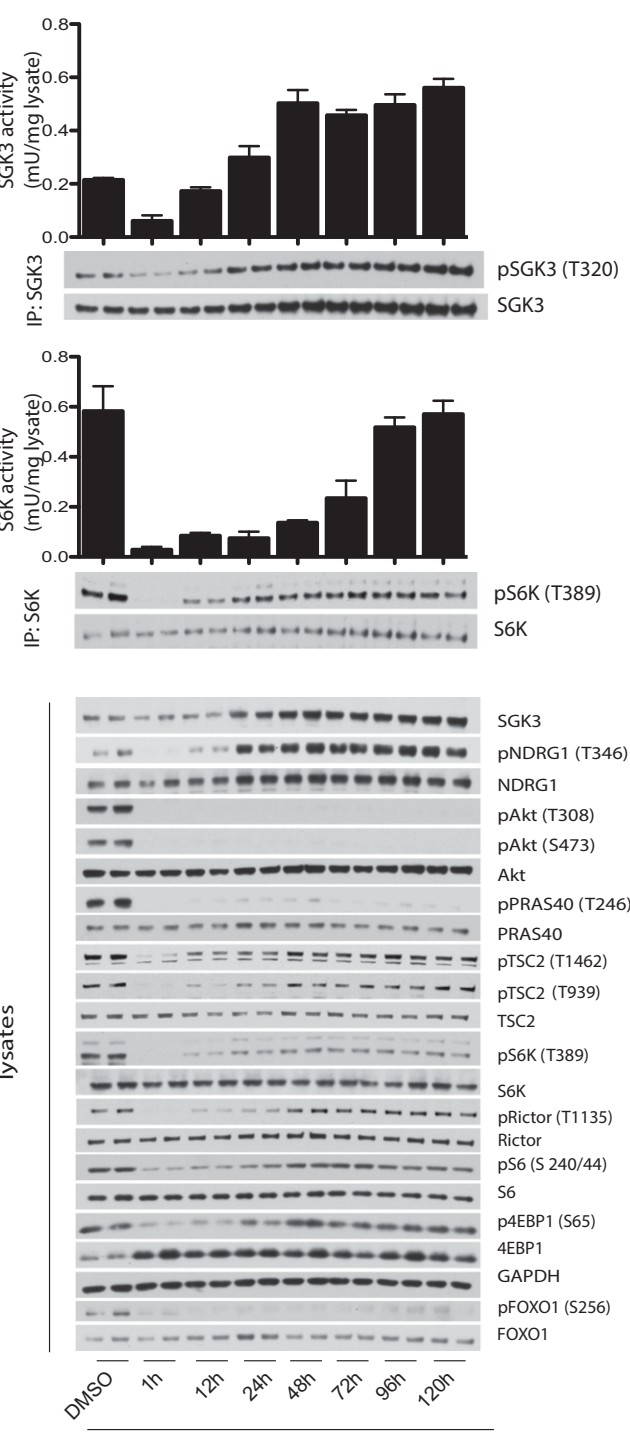

**Figure 6. SGK3 counteracts inhibition of the PI3K/Akt pathway by phosphorylating TSC2 and stimulating S6K1.**

ZR-75-1 cells were treated with 1 μM MK-2206 for the indicated times. SGK3 (upper panel) and S6K1 (middle panel) were immunoprecipitated and subjected to *in vitro* kinase assay by measuring phosphorylation of the Crosstide substrate peptide in the presence of 0.1 mM [γ-$^{32}$P]ATP in a 30 min 30°C reaction. Kinase reactions are presented as means ± SD for triplicate reaction. Immunoprecipitates (IP) were also analysed by immunoblot with the indicated antibodies. The cell lysates were also analysed by immunoblot using the indicated antibodies (lower panel).

    

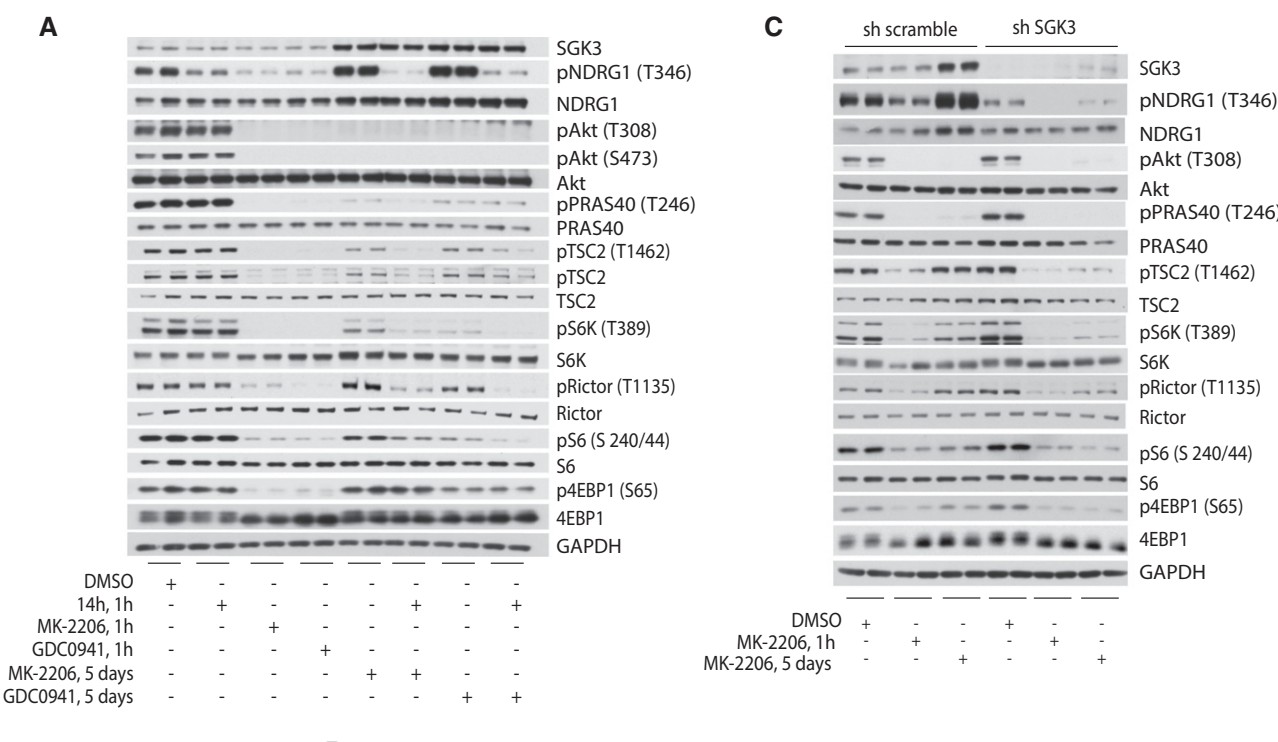

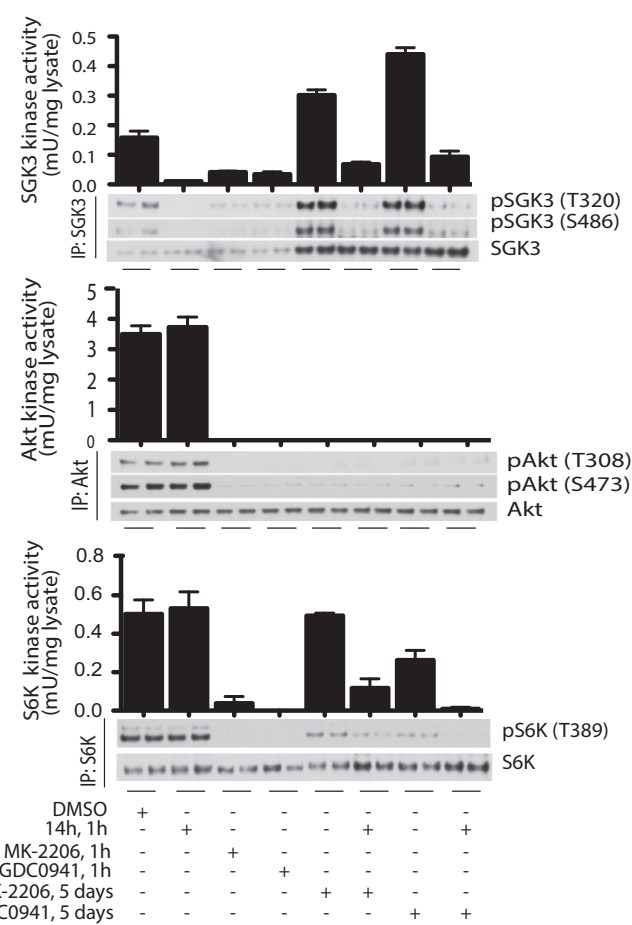

**Figure 7.**

**Figure 7.  SGK3 counteracts inhibition of the PI3K/Akt pathway by phosphorylating TSC2 and stimulating mTORC1.**
ZR-75-1 cells were treated for 1 h or 5 days with 1 μM MK-2206, 1 μM GDC0941 or 3 μM 14h inhibitors, alone or in combination, as indicated.

A  The cell lysates were analysed by immunoblot using the indicated antibodies.

B  SGK3 (upper panel), Akt1 (middle panel) and S6K1 (lower panel) were immunoprecipitated from the same cell lysates and subjected to *in vitro* kinase assay by measuring phosphorylation of the Crosstide substrate peptide for kinases in the presence of 0.1 mM [γ-$^{32}$P]ATP in a 30 min, 30°C reaction. Kinase reactions are presented as means ± SD for triplicate reaction. Immunoprecipitates (IP) were also analysed by immunoblot with the indicated antibodies.

C  SGK3 was knocked down in ZR-75-1 cells by using shRNA probe B and compared to a control shRNA probe, named sh scramble. After infection, the cells were kept for 2 days in puromycin selection media and then seeded for the experiment. The cells were treated with 1 μM MK-2206 for 1 h or 5 days. The cell lysates were subjected to immunoblot analysis with the indicated antibodies.

(Sommer *et al*, 2013) and in which prolonged treatment with Akt inhibitors leads to upregulation of SGK3 (Fig 8A). Measurement of cell confluency over 5–6 days in the continued presence of the Akt inhibitors, MK2206 or AZD5363, revealed a dose-dependent inhibition of cell growth compared to DMSO-treated cells (Fig 8B upper and bottom panel and Fig EV4A and B). Treatment with SGK inhibitor (14h) alone had little anti-proliferative effect, even at 3 μM concentration (Fig 8B upper and bottom panel and Fig EV4A and B). To determine whether a combination effect could be observed following prolonged inhibition of Akt, a long-term growth assay was undertaken. Culture in the presence of 0.3 μM Akt inhibitors (AZD5363 or MK-2206) dramatically slowed the growth of BT-474c cells but they eventually reached confluency after approximately 24 and 20 days, respectively. Addition of 3 μM SGK inhibitor (14h) induced further reduction in cell growth when it was combined with Akt inhibitor (AZD5363 or MK-2206) from the beginning of treatment or added sequentially 12 days after initial administration of Akt inhibitor (AZD5363 or MK-2206) alone (Fig 8B, upper and bottom panel).

The anti-tumour efficacy of the Akt (MK-2206) and SGK (14h) inhibitors as monotherapy and in combination was investigated in the BT-474 human breast xenograft model. At 100 mg/kg MK-2206, the *in vivo* growth of BT-474 was inhibited at ~20% (*P* < 0.01); however, no regression as monotherapy was observed. No anti-tumour effects were observed following treatment with the SGK inhibitor alone (14h, 25 mg/kg). However, concomitant administration of both agents had a significantly greater effect than that observed with either agent alone, including ~80% regression in all tumours (*P* < 0.001). No toxicity or weight loss relative to the vehicle control group was observed in any treatment group (Fig 8D). The concentration of inhibitors in the plasma measured 2–3 h after

the final administration was 1–3 μM (Fig 8E), without significant differences when administered as a monotherapy or combination. To evaluate whether tumour regression in combination treatment was due to apoptosis, we performed immunohistochemical analysis of tumour samples with anti-cleaved caspase-3 antibodies. Results showed increased number of apoptotic cells in Akt inhibitor (MK-2206) monotreatment as compared to vehicle treatment, and the number of apoptotic cells was significantly higher (*P* < 0.001) in combination (MK-2206 and 14h) treatment compared to MK-2206 treatment alone (Fig 8F). Taken together, these data indicate that inhibition of both Akt and SGK is required to achieve superior anti-tumour activity in these xenografts.

To elucidate possible mechanisms for anti-proliferative effect with SGK and Akt inhibitor combination treatment in *in vitro* and *in vivo* experiments, we analysed tumour samples taken at the end of the treatment. Immunohistochemical and immunoblot analysis revealed that Akt inhibitor (MK-2206) alone ablated Akt 473 and PRAS40 phosphorylation and partially reduced NDRG1 phosphorylation and S6 protein phosphorylation (Fig 8G and H). SGK inhibitor (14h) when administered alone had no major effect on phosphorylation of any of these markers (Fig 8G and H). Combination of Akt and SGK inhibitors (MK-2206 and 14h) resulted in the ablation of NDRG1 phosphorylation and a more moderate inhibition of S6 protein phosphorylation than was observed with MK-2206 inhibitor alone (Fig 8G and H). Immunoblot analysis also revealed that SGK3 protein level was not markedly upregulated in tumours of mice treated with either Akt inhibitor (MK-2206) alone or in combination with SGK3 inhibitor (14h). However, the phosphorylation of NDRG (Thr246) was not ablated in Akt inhibitor treatment, indicating higher SGK3 activity, whereas combination treatment (MK-2206 and 14h) induced marked dephosphorylation

**Figure 8.  Dual treatment with Akt and SGK inhibitors reduces tumour growth in BT-474 xenograft model.**

A  BT-474 cells were treated for the indicated times with 0.3 μM MK-2206. The cell lysates were analysed by immunoblot using the indicated antibodies.

B  BT-474c cells were treated with inhibitors as indicated either as monotherapy or in combination and cell confluency measured on the Incucyte ZOOM every 4 h for up to 4 weeks.

C  BT-474 cells were injected subcutaneously into athymic *Foxn1$^{nu}$* nude mice. Mice were treated with either vehicle (8 mice) or MK-2206 (100 mg/kg) (10 mice) or 14h (25 mg/kg) (6 mice) or both, MK-2206 and 14h (10 mice) for 24 days. The tumour volume was measured twice a week. Tumour growth is represented as the fold change mean ± SEM.

D  All mice were weighed at the end of the treatments. Results are presented as mean ± SD.

E  Plasma concentrations of MK-2206 and 14h were analysed in samples obtained 2–3 h after the administration of the last dose on the 24$^{th}$ day of treatment. Results are presented as a mean ± SD from three different samples.

F  Tumours were harvested at the end of the experiment, 4 h after the last dosage and subjected to immunohistochemical analysis using cleaved caspase-3 antibody (clCasp3). Apoptotic cells were counted in 25 fields per condition (left panel) and quantified as clCasp3-positive cells/field (right panel). The results are presented as mean ± SD number of cleaved caspase-3 positive cells. Representative images are shown. Scale bar, 100 μm.

G  Tumours were harvested at the end of the experiment, 4 h after the last dosage and subjected to immunohistochemical analysis with the indicated antibodies. Representative images are shown. Scale bar, 100 μm.

H  Tumours were obtained the same way as in (G) and subjected to immunoblot analysis with the indicated antibodies. Six different tumours were analysed from each treatment group and each line represents one tumour sample.

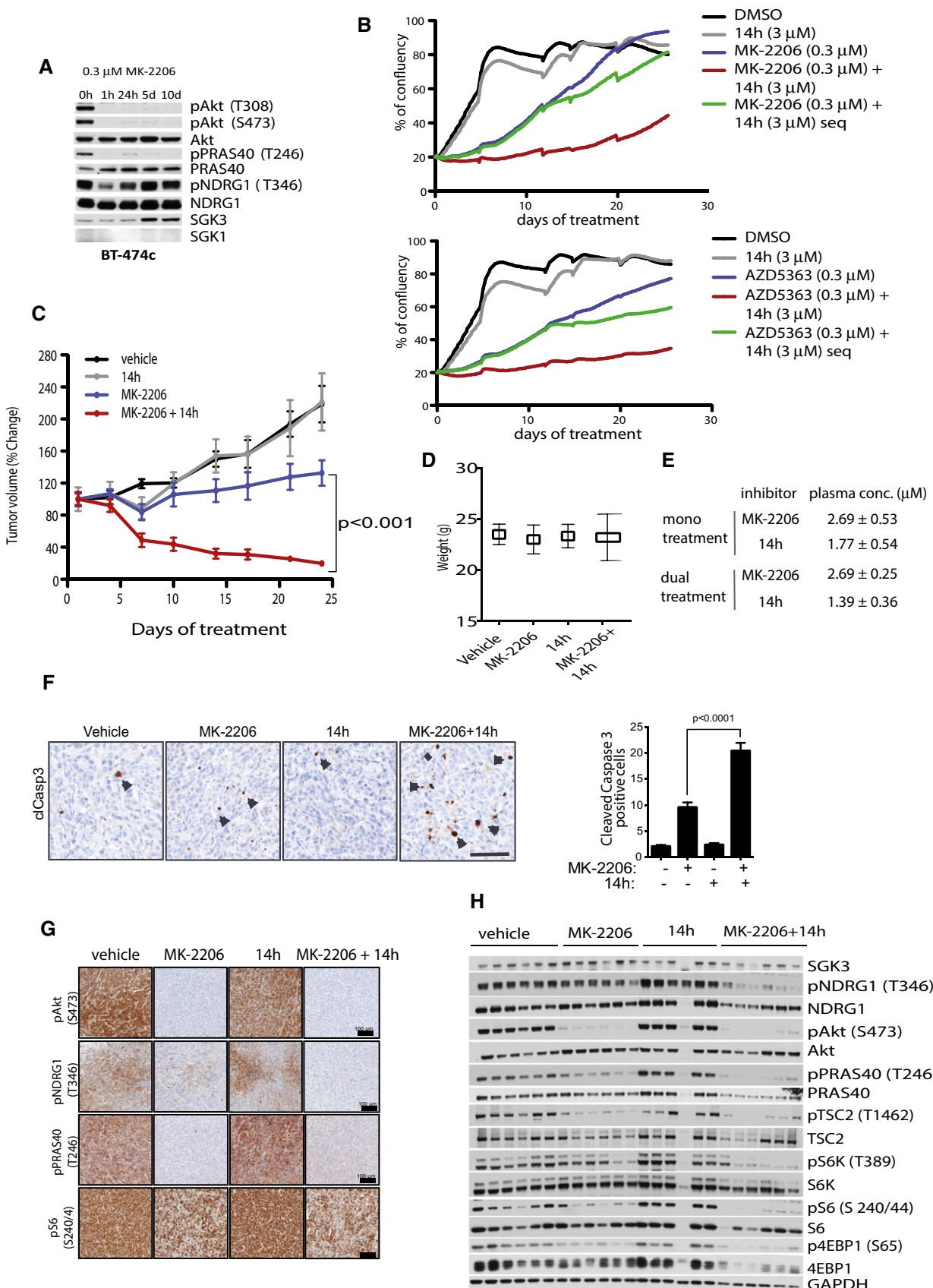

**Figure 8.**

of NDRG1 (Fig 8H). Additionally, phosphorylation of S6K1 (Thr389), S6 (Ser240/244) and 4EBP1 (Ser65) was detectable in samples of mice treated with Akt inhibitor (MK-2206) alone, whilst they were severely diminished in samples of mice receiving combination treatment (Fig 8H). These results indicate possible reactivation of the mTORC1 pathway after monotreatment with the Akt inhibitor (MK-2206). However, it is not clear whether the reactivation was mediated through SGK3 phosphorylating TSC2 at Akt sites (Thr1462), since the tumour samples obtained from monotreated (MK-2206) or combination (MK-2206 and 14h) mice showed similar level of TSC2 phosphorylation (Fig 8H). Further analysis of BT-474c cell line treated for 5 days with Akt inhibitor (MK-2206) and subsequent 1-h treatment with SGK3 inhibitor (14h) showed the similar results as observed in ZR-75-1 cells. We observed suppression of phosphorylation of TSC2, S6K1 and its downstream target S6 protein (Fig EV4C). Phosphorylation of 4EBP1 was not markedly reduced upon SGK3 inhibition (Fig EV4C). SGK inhibitor (14h) did not significantly suppress phosphorylation of TSC2, S6K1, S6 protein or 4EBP1 in BT-474c cells cultured in serum in the absence of prolonged treatment with Akt inhibitor (Fig EV4C).

**Exploitation of digital barcoding technology to quantify human kinome mRNA after inhibitor exposure**

Given the profound effects of prolonged treatment with Akt and Class I PI3K inhibitors on SGK3 mRNA in cells, we next quantified the effects of MK-2206, AZD5363 and GDC0941 across the complete human protein kinase superfamily (Table EV1). To accomplish this, we employed NanoString technology (Geiss *et al*, 2008) to digitally capture dynamics of the kinase transcriptome in response to all 3 compounds after 5 days of compound exposure (Fig 9). These data revealed complex reprogramming signatures and the appearance of compound-specific profiles. For example, exposure to MK-2206 (Fig 9A) and AZD5363 (Fig 9B) both promoted an increase in RET and SGK110 (both from low basal levels), and a marked decrease in CDK6 mRNA levels. In contrast, exposure to the PI3K inhibitor GDC0941 (Fig 9C) induced a more generalised kinome-wide increase in mRNA levels alongside a profound elevation in SGK3 mRNA. Indeed, out of the 536-member kinase panel, SGK3 mRNA had increased to become the 48[th] highest level of expression by day 5, compared to 230[th] in the matched DMSO control, 94[th] in MK-2206 and 144[th] in AZD5363. In contrast, we found that PKCγ mRNA levels consistently decreased after exposure to all 3 inhibitors. These data revealed that although SGK1 and SGK2 were expressed at extremely low levels in ZR-75-1 cells (Table EV1), very small drug-induced increases in SGK2 could be detected using a sensitive digital assay (Table EV1). However, SGK2 increases were very low compared to SGK3 (Table EV1), which had reached an expression level equivalent to ~0.5% of the entire kinome mRNA after GDC0941 exposure for 5 days.

The six human AKT and SGK protein kinases belong to the AGC kinase subfamily. To evaluate specific effects of Akt/PI3K inhibitors on the complete AGC family, we quantified actual and relative levels of the 60 human AGC kinases in triplicate samples after exposure to compounds for either 2 days (Appendix Fig S3) or 5 days (Appendix Fig S4). SGK3 was consistently the most highly upregulated AGC kinase mRNA after exposure to all 3 compounds.

Interestingly, the Akt inhibitors AZD5363 (2- to 3-fold upregulation) or MK-2206 (4- to 5-fold upregulation) were less potent inducers of SGK3 mRNA compared to GDC0941 (6- to 10-fold increase). Several other AGC kinases, including RSKL2 and MSK1, also demonstrated statistically significant upregulation in response to several of these compounds (Appendix Figs S3 and S4). The AGC kinome data set also provided the opportunity to assess relative absolute mRNA levels. Interestingly, AKT1 and AKT2 mRNA levels were the highest amongst all AGC kinases in ZR-75-1 cells under the conditions studied, consistent with a key endogenous pro-survival role. Consistent with our previous findings, exposure of ZR-75-1 cells to these inhibitors for either 2 or 5 days (Appendix Figs S3 and S4) had a specific and profound effect on SGK3 mRNA content. For example, SGK3 mRNA represented only the 29[th] most highly expressed AGC kinase in the presence of DMSO vehicle, but it had increased to become the 12[th] most highly abundant AGC kinase mRNA after exposure to MK-2206 and the 7[th] most highly abundant after exposure to GDC0941.

## Discussion

Akt isoforms play crucial roles in regulating survival and proliferation of cancer cells, and it would therefore be expected that loss of Akt activity resulting from prolonged treatment with Class I PI3K or Akt inhibitors would not be well tolerated. Such tumours would thus be under great therapeutic pressure to upregulate additional signalling pathways to compensate for this loss of Akt activity. The ability to rapidly upregulate the SGK3 pathway represents an ingenious solution not only because SGK3 possesses similar substrate specificity as Akt and hence the ability to phosphorylate at least a subset of overlapping substrates but also because SGK3 can be activated independently of Class I PI3K via hVps34. Therefore, upregulation of SGK3, which we observed in our models of Akt-sensitive breast cancer cell lines, could serve as a strategy to circumvent inhibition of Class I PI3K as well as Akt isoforms and provide adaptive resistance response. Indeed, immunoblot analysis of ZR-75-1 total cell extracts employing the p-Akt motif antibody suggested that 6 out of the 9 detected Akt substrates are likely to be phosphorylated by both Akt and SGK3 (Fig EV3C).

By virtue of its PX domain, SGK3 is capable of binding to PtdIns(3)P produced by hVps34 on the endosomes (Bago *et al*, 2014) and our biochemical data suggest that PtdIns(3)P binding can directly promote phosphorylation and activation of SGK3 by PDK1 (Fig 4). However, it has also been reported that metabolism of PtdIns(3,4,5)P$_3$ to PtdIns(3)P at the plasma membrane via the consecutive actions of the SHIP2 and INPP4A/B inositol phosphatases can activate SGK3, and these conclusions are partially based on the findings that Class I PI3K inhibitors suppress SGK3 activity (Fig 2B) (Gasser *et al*, 2014; Chi *et al*, 2015). SGK3 is also partially suppressed by treatment with Class I PI3K inhibitors in U2OS cells (Bago *et al*, 2014). In our opinion, it has not yet been possible to rule out whether the effect of Class I PI3K inhibitors on SGK3 activity is a consequence of these compounds inhibiting the activation of mTORC2 that triggers SGK3 activation by phosphorylating the hydrophobic motif rather than an effect on plasma membrane PtdIns(3)P levels generated via metabolism of PtdIns (3,4,5)P$_3$.

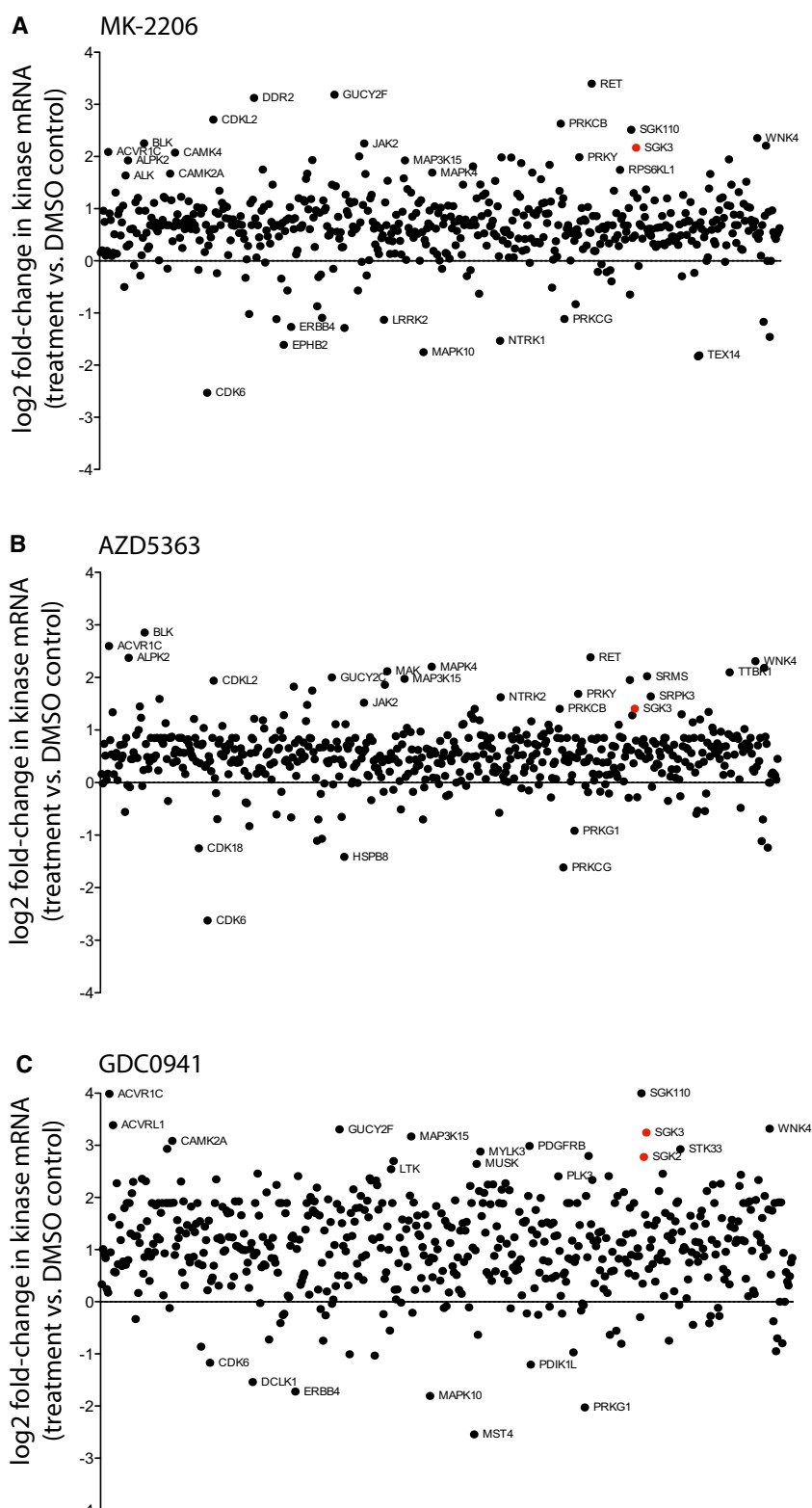

**Figure 9.  Prolonged exposure to Akt or Class I PI3K inhibitors induces transcriptional changes in human kinome.**

A–C    ZR-75-1 cells were treated for 5 days with 1 μM MK-2206 (A), 1 μM AZD5363 (B), 1 μM GDC0941 (C) or DMSO. Human mRNAs were hybridised to NanoString human kinome and control code sets, then subjected to quantification using NanoString software. Results are presented as mRNA change of each kinase relative to mRNA isolated from control sample treated with DMSO. To permit data compaction, and simple kinome-wide comparisons, the fold changes are log2 transformed. The kinase mRNAs exhibiting prominent changes are annotated, with SGK3 highlighted in red. Similar profiles were obtained in second independent experiment.

Other SGK isoforms (SGK1 or SGK2) do not possess a PX domain and, like Akt, are activated downstream of Class I PI3K isoforms. It would therefore be unlikely that SGK1 and SGK2 could substitute for Akt in cells treated with Class I PI3K inhibitors, which may explain why we did not observe marked upregulation of SGK1 or SGK2 in the experiments we undertook (Figs 1 and 2). The conserved emergence of SGK3 as a key inducible resistance determinant might potentially be explained by the very high endogenous levels of AKT1 and AKT2 expression already present in these cells, and the almost complete absence of AKT3, SGK1 and SGK2 mRNAs (Table EV1) as potential compensatory mechanisms for when Akt1 and Akt2 activity are blocked.

The SGK3 gene has been reported to be an oestrogen or androgen receptor transcriptional target in ER-positive breast cancer cells (Wang *et al*, 2011; Xu *et al*, 2012) and in androgen receptor (AR)-positive prostate cancer (Wang *et al*, 2014). It would be important to investigate the molecular mechanism by which prolonged inhibition of Akt or Class I PI3K leads to kinome-wide changes in mRNA expression levels, and in particular SGK3 mRNA and whether the upregulation of SGK3 protein and activity is limited to ER/AR cell backgrounds/tumours, or applicable to broader group of cancer cells. It would also be interesting to perform kinome-wide mRNA profiling in a wide selection of tumour or patient-derived cells before and after inhibitor exposure, so that drug resistance signatures in addition to SGK3 can be compared and contrasted as potential guides to inform targeted therapeutics.

The notion that SGK isoforms can play an important role in controlling growth and proliferation is also supported by research undertaken in model organisms such as *Caenorhabditis elegans* and budding yeast. Studies in *Caenorhabditis elegans* reveal that SGK rather than Akt is the key mediator of proliferation responses by controlling fat metabolism, reproduction and lifespan (Jones *et al*, 2009; Soukas *et al*, 2009). In budding yeast, the SGK homologues termed YPK1 and YPK2 also play a vital role in regulating metabolic and other responses required for viability and proliferation (Casamayor *et al*, 1999; Niles *et al*, 2012). Previous work has shown that SGK3 is overexpressed in a variety of cancer cell lines and knockdown of SGK3 in these cells including in ZR-75-1 cells employed in this study has substantial impact on proliferation (Virbasius *et al*, 2001; Gasser *et al*, 2014; Chi *et al*, 2015), results we have been able to confirm (RB, data not shown).

Our data reveal that by phosphorylating TSC2 and activating mTORC1, SGK3 can mediate what has previously been referred to as "Akt-independent signalling" (Bruhn *et al*, 2010, 2013). It is likely that in addition to TSC2, SGK3 will phosphorylate a group of Akt substrates that could play important roles in driving metabolic, transcription and translational responses needed for the survival and proliferation of cancer cells. Consistent with this, 6 out of the 9 Akt substrates detected by immunoblot analysis of ZR-75-1 cell extracts appear to be phosphorylated by both Akt and SGK3 (Fig EV3C). However, it should be noted that SGK3 does not phosphorylate PRAS40 (Thr246) or Foxo1 (Ser256) (Fig 6), indicating that a subset of Akt substrates are not phosphorylated by SGK3. In these studies, we did not detect any SGK3 selective substrates that are not phosphorylated by Akt (Fig EV3C). Further work would be required to understand the mechanisms determining which of the Akt substrates can be phosphorylated by SGK3, as this could be controlled by consensus motif sequences and/or cellular localisation of the substrate.

We also noticed that under conditions of sustained Akt inhibition in ZR-75-1 cells, in which SGK3 is activated, addition of SGK3 inhibitor (14h) (Fig 7A), whilst suppressing S6K1 phosphorylation, failed to commensurately reduce 4EBP1 phosphorylation. Similar results have been observed in other cell lines treated with the mTORC1 inhibitor rapamycin or starved from amino acids, where S6K1 phosphorylation was suppressed to a much greater extent than 4EBP1 (Choo *et al*, 2008; Kang *et al*, 2013). Other mTORC1 substrates have also been reported to be differentially affected following inhibition of mTORC1 (Kang *et al*, 2013). It has been suggested that this results from mTORC1 substrates being deferentially sensitive to residual mTORC1 activity depending on the type and the duration of the stress conditions (Kang *et al*, 2013). We have also observed that the hVps34 inhibition leads to a greater dephosphorylation of 4EBP-1 than seen with SGK3 inhibition (Fig EV3B). This could be accounted for if the inhibition of hVps34 led to suppression of mTORC1 through mechanisms additional to SGK3. Indeed, several studies report regulation of mTORC1/S6K1 by hVps34 in response to amino acid starvation through an ill-defined mechanism (Byfield *et al*, 2005; Nobukuni *et al*, 2005). We also observed that in BT-474c cells, the inhibition of SGK3 induced a more moderate inhibition of S6K1 than in ZR-75-1 cells (Figs 6 and EV4C), indicating that there is likely to be variation between cancer cell lines.

In this study, we have further characterised 14h, a novel small-molecule cell-permeant SGK inhibitor initially reported by Sanofi (Halland *et al*, 2015). 14h and the other related inhibitors we have analysed (14g, 14i and 14n) are much more specific than a previously reported and widely used SGK inhibitor GSK650394 (Sherk *et al*, 2008) that inhibits many kinases other than SGK1 (http://www.kinase-screen.mrc.ac.uk/screening-compounds/34880 7?order = field_results_inhibition&sort = asc). The only other relatively selective SGK1 inhibitor that we are aware of, EMD638683 (http://www.kinase-screen.mrc.ac.uk/screening-compounds/61759 4), is not potent displaying an *in vitro* IC$_{50}$ of only 3 μM (Ackermann *et al*, 2011). Consistent with this, extremely high doses of 100 μM EMD638683 are required to observe marginal reduction in NDRG1 phosphorylation in breast cancer cells that express high levels of SGK1 (Eeva Sommer, data not shown). Our data demonstrate that 1–3 μM 14h optimally suppresses SGK3-mediated phosphorylation of NDRG1 and TSC2 in cells (Figs 5–7). We also observed that 14h in addition to inhibiting SGK3 catalytic activity also suppressed phosphorylation of both the T-loop and hydrophobic motif of SGK3 in cells (Fig 5E), as well as inhibiting PDK1 from phosphorylating SGK3[S486E] in the presence of PtdIns(3)P *in vitro* (Fig 5F). This is analogous to Akt allosteric inhibitors such as MK-2206 that interact in an interface between the PH domain and the kinase domain, trapping Akt in a conformation that cannot be activated by PDK1 and mTORC2 (Green *et al*, 2008; Calleja *et al*, 2009; Wu *et al*, 2010). 14h was expected to function as a conventional Akt competitive inhibitor (Halland *et al*, 2015), so it is unusual for this compound to also work by preventing the activation of SGK3 by its upstream activators.

The finding that the growth of BT-474 breast cancer cells in *in vitro* and in a xenograft model is highly sensitive to a combination of Akt (MK-2206) and SGK (14h) inhibitors that induced inhibition of cell growth and tumour regression emphasises the

therapeutic potential of a strategy of targeting both the Akt and SGK kinases for the treatment of cancer. We cannot rule out that 14h has off-target effects in a xenograft model and it would be critical to repeat these studies with a structurally diverse SGK inhibitor when it becomes available. Also, it would be useful to analyse the sensitivity of a broader number of tumours to combinations of SGK and Akt or Class I PI3K inhibitors.

In conclusion, our findings highlight the critical importance of the hVps34-SGK3 pathway and reveal that this upregulated signalling module represents a major mechanism that cells utilise to counteract inhibition of the PI3K/Akt signalling network. SGK3 is consistently the most highly upregulated kinase mRNA amongst the 60 human AGC kinases evaluated in ZR-75-1 cells exposed to either Class I PI3K or Akt inhibitors. Our findings also reveal the hVps34-SGK3 signalling pathway operates as a PI3K/Akt-independent network to stimulate mTORC1 and likely other pathways that promote cancer growth. We demonstrate that sustained suppression of Class I PI3K or Akt activity over several days triggers the selective upregulation of SGK3 mRNA and protein as well as its catalytic activity in a variety of breast cancer cell lines. Our results suggest that SGK3 is specifically activated by the hVps34 lipid kinase that generates PtdIns(3)P on endosomal membranes. PtdIns(3)P then binds to the PX domain of SGK3, promoting its phosphorylation and activation by PDK1. Our findings indicate that once activated in cells that lack Akt activity, SGK3 substitutes for Akt by phosphorylating TSC2 to trigger activation of mTORC1. Our characterisation of the 14h SGK inhibitor suggests that it will become a useful research tool to probe signalling responses controlled by SGK isoforms in vivo. 14h represents a valuable addition to our growing armoury of signal transduction inhibitors to dissect functional roles of protein kinases. Finally, we establish that combined administration of Akt (MK-2206) and SGK (14h) inhibitors to mice bearing xenograft BT-474 breast cancer tumours induces a striking tumour regression compared to when the inhibitors were administered individually. These findings highlight the therapeutic potential of a strategy targeting both the Akt and SGK kinases for the treatment of cancer.

# Materials and Methods

## Materials

Protein-G Sepharose, Glutathione-Sepharose (Amersham Biosciences); [γ-$^{32}$P]ATP (Perkin Elmer); Triton X-100, EDTA, EGTA, sodium orthovanadate, sodium glycerophosphate, sodium fluoride, sodium pyrophosphate, 2-mercaptoethanol, sucrose, benzamidine, Tween-20, Tris–HCl, sodium chloride, magnesium acetate, DMSO, reduced glutathione (Sigma); phenylmethylsulfonyl fluoride (PMSF) (Melford); tissue culture reagents, Novex 4–12% Bis-Tris gels, NuPAGE LDS sample buffer 4× (Invitrogen); Ampicillin (Merck); P81 phosphocellulose paper (Whatman); methanol (VWR Chemicals). liver phosphatidylcholine, brain phosphatidylserine, 1,2-dioleoyl-sn-glycero-3-(phosphoinositol-3-phosphate) and 1,2-dioleoyl-sn-glycero-3-phosphoinositol (Avanti Polar Lipids). Inhibitors GDC0941 (Axon Medchem), BKM120 (ChemieTek), MK-2206 (Selleck), GSK2334470 (Tocris), AZD8055 (Selleck) were purchased from the indicated suppliers. AstraZeneca provided

AZD5363. VPS34-IN1 was synthesised as described in Bago et al (2014). SAR405 (CAS: 1523406-39-4) was synthesised as described in Ronan et al (2014). Sanofi compounds 14g, 14h, 14i and 14n were synthesised as described previously (Halland et al, 2015). All recombinant proteins, DNA constructs, antibodies and inhibitors including VPS34-IN1 and SAR405 generated for the present study are described and can be requested on our reagents website (https://mrcppureagents.dundee.ac.uk/).

## General methods

DNA procedures were undertaken using standard protocols. DNA constructs were purified from E. coli DH5alpha using maxi prep kit (Qiagen). DNA sequence of the DNA constructs used in this study was performed by the Sequencing Service (MRC Protein Phosphorylation Unit, College of Life Sciences, University of Dundee, UK; www.dnaseq.co.uk).

## Antibodies

The following antibodies were raised in sheep, by the Division of Signal Transduction Therapy (DSTT) at the University of Dundee, and affinity purified against the indicated antigens: anti-Akt1 (S695B, third bleed; raised against residues 466–480 of human Akt1: RPHFPQFSYSASGTA), anti-NDRG1 (S276B third bleed; raised against full-length human NDRG1) (DU1557), anti-SGK3 (S037D second bleed; raised against human SGK3 PX domain comprising residues 1–130 of SGK3) (DU2034), anti-PDK1 (S682, third bleed; raised against residues 544–556 of human PDK1: RQRYQSHP-DAAVQ) and anti-S6K1 (S417B, 2$^{nd}$ bleed; raised against residues 25–44 of human S6K1: AGVFDIDLDQPEDAGSEDEL). Anti-phospho-Akt Ser473 (#9271), anti-phospho-Akt Thr308 (#4056), anti-phospho-NDRG1 Thr346 (#5482), anti-GAPDH (#2118), anti-phospho-TSC2 Thr1462(#3617), anti-phospho-TSC2 Ser939 (#3615), anti-TSC2 (#3612), anti-phospho-Rictor Thr1135 (#3806), anti-Rictor (#2140), anti-phospho-S6K1 Thr389 (#9205), anti-phospho-rpS6 Ser240/244 (#2215), anti-phospho-rpS6 Ser235/36 (#4856), anti-rpS6 (#2217), anti-phospho-4EBP1 Ser65 (#9451), anti-4EBP1 (#9452), anti-phospho-SGK3 Thr320 (#5642) and anti-phospho-Akt substrate (RxRxxS/T) (#10001) antibodies were purchased from Cell Signalling Technology. Anti-(phospho-SGK hydrophobic motif [Ser486 in SGK3]) antibody (#sc16745) was from Santa Cruz Biotechnology, and total anti-SGK antibody was from Sigma (#5188). Secondary antibodies coupled to HRP (horseradish peroxidase) were obtained from Thermo Scientific.

## Cell culture and cell lysis

ZR-75-1, CAMA-1, T47D and BT-474c cell lines were sourced as described previously (Davies et al, 2012). HEK293 cells were purchased from the American Tissue Culture Collection (ATCC). Cells were cultured in RPMI or DMEM media supplemented with 10% (v/v) foetal bovine serum, 2 mM L-glutamine, 100 U/ml penicillin and 0.1 mg/ml streptomycin. Inhibitor treatments were carried out as described in figure legends. During 5 days inhibitor treatments, the media was changed after 72-h intervals. The cells were lysed in buffer containing 50 mM Tris–HCl (pH 7.5), 150 mM NaCl, 1 mM EDTA, 1 mM EGTA, 1 mM sodium

orthovanadate, 10 mM sodium glycerophosphate, 10 mM sodium pyrophosphate, 0.27 M sucrose, 0.1% (v/v) 2-mercaptoethanol, 1 mM benzamidine and 0.1 mM PMSF. Lysates were clarified by centrifugation at 16,000 $g$ for 10 min at 4°C. Protein concentration was calculated using Bradford assay (Thermo Scientific). Immunoblotting and immunoprecipitation were performed using standard procedures. The signal was developed using ECL Western Blotting Detection Kit (Amersham) on Amersham Hyperfilm ECL film (Amersham).

## Real-time PCR for SGK1, SGK2 and SGK3

Total RNA was isolated using RNA extraction kit (Qiagen), and cDNA was prepared using iSscript cDNA synthesis Kit (Bio-Rad) according to manufacturer's instructions. PCRs were done in 20 μl volume, containing 5 μl cDNA, 0.5 μM each primer (Invitrogen) and SSsoFast Eva Green Supermix (BIO-RAD). All PCRs were run as follows: 95°C for 5 min, followed by 45 cycles at 95°C for 5 s, 60°C for 30 s. Each sample was run in triplicate in three independent experiments. $2^{(-\Delta\Delta)}$ $C_t$ method was used to calculate relative mRNA expression. $C_t$ values were normalised against 18S rRNA, and relative expression was calculated using DMSO-treated cDNA sample as a calibrator. Primers used in this study were as follows: SGK1 sense: 5′CTAACGTCTTTCTGTCTC3′ and anti-sense: 5′CATAGGAGTTATTGGCAAT3′, SGK2 sense: 5′CTTCCATCTCACTAACCA3′ and anti-sense: 5′CTTTGTTATTAGGGATAGTCA3′, SGK3 sense: 5′GAAGTGAATGGTTTGTCT3′ and anti-sense: 5′ATATTCTCTTGCCAGGAA3′ and 18S sense: 5′AATGGCTCATTAAATCAGTT3′ and anti-sense: 5′CTAGAATTACCACAGTTATCC3′.

## SGK3 and Rictor knockdown and cell transfection

SGK3 knockdown was performed by using the MISSION shRNA (Sigma Aldrich) pLKO.1 vectors. Rictor knockdown shRNA sequence (insert sequence: ACCGGACTTGTGAAGAATCGTATCTTCTCGAGAAGATACGATTCTTCACAAGTTTTTTGAATTC) was cloned in pLKO1-puro vector (DU44740). Lentiviruses were produced in HEK293T cells, by transfecting 70% confluent 15-cm tissue culture dish with 10 μg DNA (8 μg shRNA-pLKO1-puro, 4 μg pCMVdelta R8.2 packaging vector (Adgene) and 2 μg pCMV-VSV-G envelope vector (Adgene)). DNA was mixed with 40 μl of polyethylenimine in 300 μl Optimem media (Invitrogen). The media containing viruses was collected and filtered 48 h after transfection. Breast cancer cell lines were kept in virus containing media for 24 h. Cells were allowed to recover in fresh media for 24 h, before the media was replaced with the selection media containing puromycin (2 μg/ml). The cells were kept in selection media for 10 days before using for the experiments, unless stated otherwise.

## Immunoprecipitation and assay of SGK3, Akt and S6K1

*In vitro* kinase activity of SGK3, Akt and S6K1 was assayed as described previously (Bago *et al*, 2014). Briefly, SGK3, Akt and S6K1 were immunoprecipitated from 1 mg ZR-75-1 cell lysate. Immunoprecipitates were washed in sequence with lysis buffer containing high salt concentration (500 mM NaCl), lysis buffer and Buffer A (50 mM Tris pH 7.5, 0.1 mM EGTA). Kinase activity was assessed by measuring [γ-$^{32}$P]ATP incorporation into Crosstide

substrate peptide [GRPRTSSFAEGKK]. The reactions were carried in 40 μl total volume containing 0.1 mM [γ-$^{32}$P]ATP (400–1,000 cpm/pmol), 10 mM magnesium acetate and 30 μM Crosstide peptide. Reactions were terminated by adding 10 μl 0.1 M EDTA and spotting 40 μl of the resulting reaction mix on P81 paper, which were immediately immersed into 50 mM orthophosphoric acid. Papers were washed at least five times in 50 mM orthophosphoric acid, rinsed in acetone and air-dried. Radioactivity was quantified by Cerenkov counting. One unit of enzyme activity was defined as amount of enzyme that catalyses incorporation of 1 nmol of [γ-$^{32}$P]ATP into the substrate over 1 min.

## Production of recombinant SGK3

SGK3 [S486E]-GST (DU 52370) and SGK3 [R90A S486E]-GST (DU52372) in pCMV5D vector were transfected in HEK293 cells using polyethylenimine. The cells were lysed 48 h post-transfection in the lysis buffer without phosphatase inhibitors following 1-h incubation in the presence of PDK1 inhibitor (GSK2334470, 5 μM) and hVps34 inhibitor (VPS34-IN1, 5 μM). The lysates were clarified by centrifugation, and GST-tagged proteins were affinity purified using GST-Sepharose beads. The proteins were eluted with 40 mM glutathione, pH 8.0, aliquoted and stored at −80°C.

## SGK3 kinase assay on phospholipid vesicles

Lipid vesicles were prepared as described in Alessi *et al* (1997b). Briefly, the mixture containing 1 mM phosphatidlycholine, 1 mM phosphatidylserine and either 0.01–0.1 mM phosphatidylinositol or 0.01–0.1 mM phosphatidylinositol 3-phosphate in chloroform was dried under vacuum and further resuspended in kinase assay buffer (10 mM Tris–HCl, pH 7.4, 150 mM NaCl, 0.1 mM EGTA) by vortexing and sonication. Multilamellar vesicles formed in this process were passed through 0.2-μm filter using lipid extruder, after which a suspension of smaller unilamellar vesicles was obtained. Solutions were stored at 4°C at concentrations 10× higher than those used in the kinase assay and used within 3 days. The kinase assay was carried out in two stages; in the first stage, 50 ng of SGK3 [S486E]-GST or SGK3 [R90A, S486E]-GST was incubated with lipid vesicles, 10 mM Mg-acetate and 0.1 mM ATP. The reaction was started by adding 50 ng of recombinant GST-PDK1 (32 U/mg) (DU954) to permit activation of SGK3-GST. After 30 min at 30°C, the reaction was stopped by adding PDK1 inhibitor (GSK2334470, 1 μM). In the second stage, the reaction mixture was topped up with 0.1 mM ATP and reaction started by adding 2 μg of recombinant GST-NDRG1 (DU1557) to permit phosphorylation by activated SGK3-GST. The assay was carried out for 30 min at 30°C. The reaction was stopped by adding 4× LDS sample buffer.

## Determination cell growth *in vitro*

For growth assays, BT-474c cells were seeded in 12-well plates at a density of $1 \times 10^5$ cells/well and left to adhere overnight. Cells were then treated with MK-2206, AZD5363 and 14h inhibitors and imaged every 4 h on the Incucyte ZOOM (Essen Bioscience) for up to 4 weeks to give a measure of cell confluency. Media was refreshed every 4–5 days. Inhibitor treatments were carried out as described in figure legends.

## Animal studies and IHC

Experiments involving mice were in accordance with Institutional Guidelines of Memorial Sloan Kettering Cancer Center (Protocol number 12-10-019). Animals were housed in air-filtered laminar flow cabinets with a 12-h light cycle and food and water *ad libitum*. BT-474 VH2 (Baselga *et al*, 1998) were resuspended in 1:1 Complete media/Matrigel (Corning) and injected subcutaneously into the flanks of 6-week-old female athymic *Foxn1^{nu}* nude mice (Harlan Laboratories). Animals' drinking water was supplemented with 1 μM 17β-E2 (Sigma). When tumours reached a volume of ~150 mm$^3$, mice were randomised, treated, and tumours were measured twice a week. Tumour volume was determined using the following formula: (length × width$^2$) ÷ 2. Tumour growth was represented as the fold change mean ± SEM. Treatments were as follows: MK-2206 (100 mg × kg$^{-1}$ in 30% Captisol (Sigma), 5 times/week, p.o.) and 14h (25 mg × kg$^{-1}$ in 40% of 3:1 Glycofurol/Kolliphor® RH 40 mixture (Sigma) in 0.9% saline, 5 times/week, p.o.). Tumours were harvested at the end of the experiment 4 h after the last dosage, fixed in 4% formaldehyde in PBS and paraffin-embedded. IHC was performed on a Ventana Discovery XT processor platform using standard protocols and the following antibodies from Cell Signalling Technology: pAkt(S473) (#4060), 1:70; pPRAS40(T246) (#2997), 1:50; pS6 (S240/4) (#5364), 1:500; pNDRG1 (T346) (#5482), 1:200; Cleaved Caspase-3 (#9664), 1:50. Primary staining was followed by 60 min incubation with biotinylated goat anti-rabbit IgG (Vector labs) 1:200. Blocker D, Streptavidin-HRP and DAB detection kit (Ventana Medical Systems) were used according to the manufacturer instructions.

## NanoString analysis

ZR-75-1 cells were collected, washed in PBS and snap-frozen. Cell pellets were resuspended at a concentration of ~6,500 cells/μl in RLT buffer (Qiagen). The equivalent of ~10,000 cells (~1.5 μl) was employed for direct kinase mRNA quantification, without the need for further mRNA purification or amplification. mRNAs were hybridised to NanoString human kinome barcode probes and control code sets, and kinase mRNA levels quantified using the nCounter colour barcoding system after count normalisation with internal housekeeping genes and 8 negative controls. For total copy numbers, triplicate normalised count values were averaged, and to calculate fold induction, means were divided by triplicate mean values from matched DMSO controls. *P*-values were calculated using unpaired *t*-test (GraphPad Prism software) by employing treated and control count data for each kinase mRNA.

## Protein kinase profiling

Protein kinase profiling against Dundee panel of 140 protein kinases was undertaken at the International Centre for Protein Kinase Profiling. The result for each kinase was presented as a mean kinase activity of the reaction taken in triplicate relative to a control sample treated with DMSO.

Assay conditions and abbreviations are available at http://www.kinase-screen.mrc.ac.uk.

## Statistical analysis

All experiments presented in this paper were performed at least twice, and similar results were obtained. Error bars indicate standard deviation.

**Expanded View** for this article is available online.

## Acknowledgements

We thank Aaron Smith (AstraZeneca, R&D, Innovative Medicines) for measuring plasma MK-2206 and 14h concentrations in mouse xenografts, and Sabina Cosulich for helpful discussion as well as members of the MRC-PPU International Centre for Kinase Profiling (coordinated by Jennifer Moran) for undertaking potency and specificity analysis of SGK inhibitors. We express gratitude to for the excellent technical support of the MRC Protein Phosphorylation and Ubiquitylation Unit (PPU) DNA Sequencing Service (coordinated by Nicholas Helps), the MRC-PPU tissue culture team (coordinated by Kirsten Airey and Janis Stark), the Division of Signal Transduction Therapy (DSTT) antibody purification teams (coordinated by Hilary McLauchlan and James Hastie). This work was supported by the Medical Research Council (MC_UU_12016/2) and the pharmaceutical companies supporting the Division of Signal Transduction Therapy Unit (AstraZeneca, Boehringer-Ingelheim, GlaxoSmithKline, Merck KGaA, Janssen Pharmaceutica and Pfizer). PAE acknowledges North West Cancer Research Fund (NWCR) for support (project grants CR1037 and CR1088).

## Author contributions

RB designed, executed experiments (Figures 2, 3, 4, 5D, 5E, 5F, 6, 7, 8H, EV1, EV2, EV3 and EV4C) , analysed data and played a major role in interpretation of data and preparation of the manuscript; ES designed, executed experiments (Figs 1, 2A and 8A) and analysed data; PC designed, executed experiments (Fig 8C, 8D, 8F, 8G), analysed data and helped with preparation of the manuscript; CC designed, executed experiments (Figures 8B, EV4A and EV4B), analysed data in Figure 8E and helped with preparation of the manuscript; FPB designed, executed experiments (Figures 9, Appendix Figure S2 and Appendix Figure S3) and analysed data; NS synthetised SGK inhibitors used in this study; JB helped with design of experiments and analysis of data; DC helped with design of experiments and analysis of data; PAE helped with design of experiments, analysis of the data and helped with the preparation of the manuscript; DRA conceived of the project, helped with experimental design and analysis and interpretation of data and played a major role in preparation of the manuscript.

## Conflict of interest

The authors declare that they have no conflict of interest.

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
