## [Review Process File · The EMBO Journal]

Manuscript EMBO-2016-93929

The hVPS34-SGK3 pathway alleviates sustained PI3K/Akt inhibition by stimulating mTORC1 and tumour growth

Ruzica Bago, Eeva Sommer, Pau Castel, Claire Crafter, Fiona P. Bailey, Natalia Shpiro, José Baselga, Darren Cross, Patrick A. Eyers and Dario R. Alessi

Corresponding author: Dario Alessi, University of Dundee

Review timeline:

Submission date:	21 January 2016
Editorial Decision:	26 February 2016
Revision received:	23 May 2016
Editorial Decision:	14 June 2016
Revision received:	19 June 2016
Accepted:	04 July 2016

Transaction Report:

Editor: Daniel Klimmeck

1st Editorial Decision

26 February 2016

Thank you for the submission of your manuscript entitled 'The hVPS34-SGK3 pathway counteracts inhibition of the PI3K-Akt to stimulate mTORC1 and tumour growth' (EMBOJ-2015-93929) to The EMBO Journal. Your study has been sent to three referees, and we have received reports from all of them, which I copy below.

As you will see, all referees acknowledge the potential high interest and novelty of your work, although they also express a number of specific major concerns that would need to be addressed before they can support publication of your manuscript in The EMBO Journal. In particular, referee #1 points out the need for you to prove the direct dependency of MK-2206 induced mTORC1 signaling on 14h activity (ref #1). Accordingly, referee #1 sees some of the conclusions not sufficiently well supported by the experimental data provided. In line, referees #2 and 3 ask for revisions regarding the inhibitor experiments (ref #2 pt.1; ref #3 pt.1,3). I judge the comments of the referees to be generally reasonable, thus we are in principle happy to invite you to revise your manuscript experimentally to address the referees' comments.

Please contact me if you have any questions, need further input on the referee comments or if you anticipate any problems.

Please be aware that it is 'The EMBO Journal' policy to allow a single round of revision only and that, therefore, acceptance of the manuscript will essentially depend on the completeness of your responses included in the next version of the manuscript.

We generally allow three months as standard revision time in the first instance. As a matter of policy, competing manuscripts published during this period will not be taken into consideration in our assessment of the novelty presented by your study ("scooping" protection). Nevertheless, please contact me as soon as possible upon publication of any related work in order to discuss how to proceed. Should you foresee a problem in meeting this three-month deadline, please let us know in advance and we may be able to grant an extension.

When preparing your letter of response to the referees' comments, bear in mind that this will form part of the Review Process File, and will therefore be available online to the community. For more details on our Transparent Editorial Process initiative, please visit our website:

http://emboj.msubmit.net/html/emboj_author_instructions.html#a2.12

As you have probably seen already, every paper now includes a 'Synopsis', displayed on the html and freely accessible to all readers. The synopsis includes a 'model' figure as well as 2-5 one-short-sentence bullet points that summarize the article. I would appreciate if you could provide this figure and the bullet points.

Please note that as of January 2016, our new EMBO Press policy asks for corresponding authors to link to their ORCID iDs. You can read about the change under "Authorship Guidelines" in the Guide to Authors here: <http://emboj.embopress.org/authorguide>

In order to link your ORCID iD to your account in our manuscript tracking system, please do the following:

1. Click the 'Modify Profile' link at the bottom of your homepage in our system.
2. On the next page you will see a box half-way down the page titled ORCID*. Below this box is red text reading 'To Register/Link to ORCID, click here'. Please follow that link: you will be taken to ORCID where you can log in to your account (or create an account if you don't have one)
3. You will then be asked to authorise Wiley to access your ORCID information. Once you have approved the linking, you will be brought back to our manuscript system.

We regret that we cannot do this linking on your behalf for security reasons. We also cannot add your ORCID iD number manually to our system because there is no way for us to authenticate this iD number with ORCID.

Thank you very much in advance.

Finally, in order to ensure good reporting standards and to improve the reproducibility of published results, our guidelines to authors are consistent with the Principles and Guidelines for Reporting Preclinical Research issued by the NIH in 2014. Accordingly, we now require the submission of a completed author checklist, which covers in a systematic manner your practices regarding animal welfare, human subjects, data deposition, statistics and research ethics. It needs to be filled (most of the fields will not apply to your study in particular) and returned to the editorial office at revision, either via the online submission system as a supplementary file or by email (contact@embojournal.org). Please, click on the link below and follow the instructions to download the checklist file:

<http://emboj.embopress.org/authorguide>

Again, please contact me at any time during revision if you need any help or have further questions.

Thank you for the opportunity to consider your work for publication. I look forward to your revision.

REFeree REPORTS

Referee #1:

Using extensive western blot analysis and kinase assays the manuscript clearly demonstrates that prolonged AKT or PI3 kinase inhibition induces SGK3 expression and activity in several breast cancer cell lines. The studies demonstrate that activation of SGK3 is via endosomal hVps34 lipid kinase and requires both PDK1 and mTORC2. Characterization of the SGK1 inhibitor, 14h showed that this compound also inhibits SGK3. The increased SGK3 expression associated with prolonged AKT or PI3K inhibition led to phosphorylation of TSC2 and S6 kinase indicating SGK3 activates mTORC1, however 4EBP1 also down stream of mTORC1 was not phosphorylated. In a breast xenograft model MK-2206 (AKT inhibitor) decreased tumor growth whereas 14h had no effect. The combination, however, showed excellent synergy leading to a substantial decrease in tumor volume. This is an exciting result. A major shortcoming of the manuscript, however is that the authors have not shown that the decrease in the tumor volume with the combination of MK-2206 and 14h was due to 14h's ability to decrease MK-2206 induced mTORC1 signaling. Thus the biochemical data clearly showing that inhibition of SGK3 with 14h can overcome the increased signaling of mTORC1 in response to prolonged MK-2206 treatment may have nothing to do with the synergy of these agents *in vivo*. Thus the conclusion "our findings highlight the importance of the hVps34-SGK3 pathway and reveal that this signaling pathway represents a major mechanism that cells utilize to counteract inhibition of the PI3K-Akt signaling network" is not justified from the experiments outlined in this manuscript.

Major Comments

1. To support the *in vivo* data and the conclusion that SGK3 is a major mechanism to counteract inhibition of the PI3K-Akt signaling network the study requires:
 - a. Cell line data clearly demonstrating the effect of single agents as well as the combination on cell growth, proliferation and/or death. If the biochemical data translates into changes in these parameters then you would expect that there will only be synergy between MK-2206 and 14h when cells have been pretreated with MK-2206 for 5-10 days. If synergy is seen with upfront treatment (ie no MK-2206 pretreatment) then the synergy is likely not due to 14h's ability to overcome the SGK3 induced increase in mTORC1 signaling.
 - b. Assessment of tumor mTORC1 activity (eg pS6K, p-S6, P-4EBP1) and SGK3 expression/phosphorylation by western analysis is required. Measuring these parameters at an early and at a late time point would address whether increased SGK3 signaling and thus activation of mTORC1 occurs and if the synergy is due to modulation of the PI3K/mTORC1 pathway.
 - c. A second breast xenograft model should be performed to show that the synergy is not particular to the BT-474c model.
3. If MK-2206 is a specific AKT inhibitor, why does it decrease SGK3 activity? (Fig2B, Fig.6, Fig7C)
4. Why wasn't the BT-474c cell line that was used for the *in vivo* studies used in the extensive western and kinase analysis? Does MK-2206 induce mTORC1 activity in these cells and does 14h overcome this induction? These experiments should be performed.
5. The discussion is too long and poorly written. Extensive discussion on future directions that speculate how PDK1 may phosphorylate SGK3 (paragraph 2), how AKT inhibition may increase SGK3 expression (paragraph 5) or how in clinical samples you may measure increased SGK3 activity (paragraph 6) are not appropriate. Paragraph 6, 3rd line states "highly sensitive to a combination of Akt (MK-2206) and SGK (14h) inhibitors under condition where either inhibitor dispensed alone had minimal effects". MK-2206 substantially inhibited tumor growth and thus had a very good therapeutic response. So this statement is incorrect. Some points that should be included in the discussion are listed see points 3, 6, 7.

Minor:

6. The discrepancy between 14h's ability to inhibit phosphorylation of S6kinase/S6 but not 4EBP1 needs clarification.

7. Discussion around how SGK3 can modulate mTORC1 activity when it doesn't appear to phosphorylate PRAS40 the negative regulator of mTORC1 also warrants further discussion.

8. Figure 3C. Is missing an important control group - DMSO plus 1hr VPS34-IN1 treatment. The bottom labeling of this figure appears to be incorrect. There are 4 lanes for 5days of AZD5363 treatment (lanes 5,6,and 13,14). Are lanes 13,14 the missing control group?

9. Figure 7 concentrations for the inhibitors used need to added to the figure legend or figure.

10. In the text p8&9 last & first line, respectively it is stated "Consistent with SGK3 mediating phosphorylation of TSC2 and leading to the activation of mTORC1 under these conditions, we observed that treatment with 3 M 14h for 1 hour, suppressed phosphorylation of TSC2, S6K1, S6 and 4EBP1 protein (Fig 7A & 7B). From the western figure shown 14h decreased phosphorylation of TSC2, S6K1 and S6 but did not alter phosphorylation of 4EBP1.

Referee #2:

This study looks at the role of SGK3 in the development of resistance to PI3K and Akt inhibitors. SGK3 was upregulated at the mRNA, protein and activity levels within 5 days of treatment with inhibitors of Akt or class I PI3K in two breast cancer cell lines. Once induced, SGK3 was regulated by VPS34-derived PI3P and subsequent phosphorylation by PDK1 and mTORC2. It is found that a previously described SGK1 inhibitor from Sanofi (14h) also inhibits SGK3 and blocks its activating phosphorylation events. Prolonged inhibition of PI3K or Akt was found to lead to re-stimulation of TSC2 phosphorylation and subsequent re-activation of mTORC1, and this was suppressed by 14h treatment, suggesting that an SGK isoform might be capable of replacing Akt for the regulation of mTORC1. Finally, it is found that 14h treatment greatly enhances the ability of an Akt inhibitor to induce tumor regression.

This is an interesting, straightforward, and potentially clinically relevant study on the wiring and rewiring of signaling pathways key for tumorigenesis and of intense therapeutic interest. A few points of clarification are needed to support and expand the conclusions.

Specific comments:

1. Figure 5D: Regarding the assessment of off-target effects of 14h in cells, S6 phosphorylation on S235/236 is not a reliable specific readout for the activity of S6K1, the kinase with the second highest degree of sensitivity to 14h in vitro. This site can be phosphorylated by S6K2, RSK isoforms, and likely other AGC kinases. It is recommended that Rictor-T1135 phosphorylation be used as a highly specific S6K1 substrate in cells, as it is not even phosphorylated by S6K2. This ends up being an important point for interpretation of these studies and the subsequent proliferation and tumor studies.

2. Figure 7: To further confirm that these effects seen with 14h are indeed due to SGK3 inhibition, the sensitivity of TSC2 phosphorylation and mTORC1 activation to the VPS34 inhibitor should be tested, as well as the effects of RNAi-mediated knockdown of SGK3.

3. Figure 8: The Akt inhibitor appears to be primarily cytostatic in this tumor model, leading to a halt in tumor growth without measureable tumor shrinkage, and 14h co-treatment greatly reduces tumor size, suggesting cell death. To determine if these compounds are indeed synergizing to kill these tumor cells, apoptosis should be measured in the available tissues from all 4 treatment groups (e.g., IHC for cleaved caspase 3), as well as in cell culture models.

4. If frozen tumors from this study exist, SGK3 levels and more markers of mTORC1 signaling should be assessed.

Minor:

The title is overstated with regard to the comment on tumor growth. Figure 8B is the only experiment that looked at tumors, and the result does not support the claim in the title. MK-2206 prevents tumor growth in this model, so even if SGK3 is upregulated in these tumors, which was never shown, they are not growing.

Referee #3:

The manuscript by Bago et al examines the effects of long term inhibition of Akt or PI3K and demonstrates that while phosphorylation of one Akt substrate (PRAS40) remains inhibited, phosphorylation of NDRG1 bounces back to levels beyond untreated samples. They show that this is due to upregulation of SGK3 mRNA and protein, and that Vps34 (via production of PI3P), PDK1 and mTORC2 is required for SGK3 activity under these circumstances. In addition to NDRG1, phosphorylation of additional Akt substrates show a bounce back, including TSC2 (T1462 and S939), as well as the downstream mTORC1 substrates S6K (T389), 4EBP1 (S65), and S6 (S240/244). Use of a reasonably selective SGK3 inhibitor further demonstrated the requirement for this kinase in maintaining phosphorylation of the NDRG1 and TSC2 sites after long term PI3K/Akt inhibition, and the combination of an Akt inhibitor and the SGK3 inhibitor was effective in regressing BT474 xenograft tumors in immunocompromised mice.

This manuscript contains interesting and novel data that will likely be of interest to the readers of EMBO Journal. There are some major points that need to be addressed before publication, as well as some additional suggestions for clarity.

Major points:

1. Figure 1C would benefit from adding SGK3 knockdown alone to show that this kinase does not contribute to the basal phosphorylation of NDRG1. I realize this is addressed later on in the paper using VPS34 inhibitor alone (Fig. 3B) and SGK3 inhibitor alone (Fig. 5D), but to set the stage for the paper this data would be useful.
2. SFig 2A is referred to on p 6, but they mean SFig 2B
3. They should combine Akt or PI3K inhibitors with 14h at 1h to see if this is sufficient to wipe out phosphorylation of some substrates that are only partially inhibited (eg P-S6).
4. They should perform a Western blot with generic P-Akt substrate antibodies to get a sense of what proportion of Akt substrates behave like NDRG1 and TSC2, and which behave like PRAS40. (Mass spec analysis would be nice but beyond the scope of this paper). They should also discuss more what distinguishes the differential responses (localization, consensus motifs etc).
5. They should show the body weight plots for Fig. 8.

Minor points:

1. I understand why activation loop phosphorylation is affected by hydrophobic loop phosphorylation (requirement for PDK1 binding), but why is hydrophobic loop phosphorylation affected by activation loop phosphorylation (Fig. 3C).
2. What site-specific antibody are they using for P-4EBP1 and why does it increase following Akt inhibition (Fig. 6)?
3. They claim that 4EBP1 phosphorylation is suppressed by 14h following long term treatment with Akt or PI3K inhibitors, but this is not apparent to me (Fig. 7A).
4. Why is SGK3 inhibited by short term Akt inhibitor treatment?

Referee 1

1a. The Referee requested that we study the synergy between MK-2206 and 14h in cell growth assays. This has now been undertaken and the new data is presented in Fig. 8B. The results show that the growth of BT-474c cells when treated with a combination of 14h and MK-2206 is significantly reduced compared to when cells are treated with either agent alone.

1b. The Referee asked us to assess markers of mTORC1 activity in tumour samples by immunoblot analysis. This new data (Fig. 8H) demonstrates that a combination of 14h and MK-2206 significantly reduces the phosphorylation of S6K1, S6, and 4EBP1 as well as NDRG1 (the SGK3 and Akt biomarker site) compared when tumours were treated with either 14h or MK-2206 alone.

1c. The Referee requested that we perform an additional xenograph model to demonstrate synergy between 14h and MK-2206. Unfortunately, as mentioned in our letter to the Editor, this was beyond the scope of what we were capable of doing in a 3-month revision period.

2. (no point 2 from Referee included)

3. The Referee asked why we observe that MK-2206 induces a slight inhibition of SGK3 activity (Fig. 2b). We did not comment on this in the previous version of the manuscript but we have now emphasised this observation in the Results section and stated that the underlying mechanism accounting for this observation is unclear. This is seen with both MK-2206 Akt inhibitor as well as a structurally unrelated AZD5363 Akt inhibitor suggesting that this is unlikely to be an off target effect (Fig. 2b). Further work will be required to understand the mechanism lying behind this observation. It should be noted that this effect of Akt inhibitors suppressing SGK3 activity in ZR-75-1 cells is transient and is observed after 1 hour but not after 12 hours (Fig. 6).

4. The Referee asked us to demonstrate that inhibition of Akt induces SGK3 activity in BT-474c cells. We have undertaken this additional experiment and the new data is presented in Supplementary Fig. 9. The data shows that inhibition of Akt leads to up regulation of SGK3 and increased activity of mTORC1 as emphasised by enhanced phosphorylation of TSC2, S6K1, S6 protein, and 4EBP1, as well as NDRG1. These effects are suppressed with the 14h inhibitor.

5. The Referee felt that our discussion was too long and poorly written, and made several suggestions as to how this could be improved. We have extensively revised and reduced the length of our discussion, taking on board all these comments.

6. (minor) The Referee asked us to discuss the discrepancy between 14h ability to inhibit phosphorylation of S6K1 versus 4EBP1. We have now done this in the discussion and referred to several other papers in the literature where it has been shown that inhibition of mTORC1 resulting from conditions such as starvation or treatment with rapamycin results in different degrees of dephosphorylation of various mTORC1 substrates, including 4EBP1 and S6K1. David Sabatini suggested that different mTORC substrates “encode” their sensitivity for phosphorylation by residual mTORC1 activity.

7. (minor) As requested, we have now discussed how SGK3 can modulate mTORC1 activity when it doesn't appear to phosphorylate PRAS40 the negative regulator of mTORC1.

8. (minor) We have corrected the incorrect labelling on Fig. 3C.

9. (minor) We have included the concentrations of inhibitors in Fig. 7 in the figure legend.

10. (minor) The Referee requested we correct the statement (p8/9) “Consistent with SGK3 mediating phosphorylation of TSC2 and leading to the activation of mTORC1 under these conditions, we observed that treatment with 3 mM 14h for 1 hour, suppressed phosphorylation of TSC2, S6K1, S6 and 4EBP1 protein (Fig. 7A and 7B)” as SGK inhibitor altered S6 protein, but not 4EBP1 phosphorylation.

We have now corrected the statement into “Consistent with SGK3 mediating phosphorylation of TSC2 and leading to the activation of mTORC1 under these conditions, we observed that treatment with 3 mM 14h for 1 hour suppressed phosphorylation of TSC2. We also observed suppression of activity and phosphorylation of S6K1 and its downstream targets, Rictor and S6 protein (Fig 7A and 7B). However, phosphorylation of 4EBP1, other mTORC1 substrate, was not markedly reduced upon SGK3 inhibition (Fig 7A)” (page 9)

Referee 2

1. The Referee recommended that we immunoblot an S6K1 selective phosphorylation site on Rictor (Thr1135) to demonstrate that 14h does not inhibit activity of S6K1 for the data shown in Fig. 5d. We have now undertaken this additional analysis and confirmed that 14h does not suppress S6K1-mediated Rictor phosphorylation at Thr1135. We have also included Rictor Thr1135 and Rictor in immunoblot analysis in Fig. 6, 7A and C, Supplementary Fig 5, 6 and 9.

2. The Referee requested that we confirm that the effects seen with 14h in Fig. 7 are indeed due to SGK3 inhibition. To do this we have undertaken a new shRNA mediated knockdown of SGK3 expression study and demonstrate that this reduced phosphorylation of TSC2, S6K1, S6 protein, and 4EBP1, as well as NDRG1, similar to 14h. This new data is shown in Fig. 7C. The Referee also requested that we test the sensitivity of TSC2 phosphorylation and mTORC1 activation to the hVps34 inhibitor. This new data is shown in Supplementary Fig. 6. This demonstrates that hVps34 inhibition leads to suppression of TSC2, S6K1, Rictor, S6 protein, and 4EBP1.

3. The Referee requested that we study whether combination of MK-2206 and 14h lead to apoptosis in the tumour samples. To test for this, we have undertaken immunohistochemistry of cleaved Caspase 3 in tumour samples. This new data is shown in Fig. 8F and reveals that combination of MK-2206 and 14h does indeed lead to much greater apoptosis than is observed with either inhibitor administered individually.

4. We were not able to demonstrate significant up regulation of SGK3 protein level in whole tumour samples by undertaking immunoblot analysis, but nevertheless we detected a significant increase in phosphorylation of NDRG1 suggesting that the SGK3 was activated by these conditions. This data is presented in Fig. 8H.

Minor point:

We have modified our title as requested by the Referee.

Referee 3

1. The Referee requested that we perform an additional control experiment to establish whether knockdown of SGK3 alone has an effect on basal phosphorylation of NDRG1. We have performed this additional analysis which is now part of Fig. 7c. The new data demonstrate that knock down of SGK3 reduces, but does not ablate NDRG1 phosphorylation.

2. We have corrected the mistake in referring to supplementary figures and carefully re-checked all other figure citations.

3. The Referee asked us to combine Akt or Class I PI3 kinase inhibitors with 14h at a 1-hour time point to study how this effects phosphorylation of substrates such as S6 protein. This new study has been undertaken and the results are presented in Supplementary Fig. 5. This data shows that combining 14h with Akt or Class I PI3K inhibitors at 1-hour does not result in any major further inhibition of phosphorylation of these substrates (the substrates we have measured are pTSC2, pS6K1, pRictor, pS6, p4EBP1 and, as well as pNDRG1).

4. The Referee asked us to perform immunoblot analysis with the generic p-Akt motif substrate antibody to gauge what proportion of Akt substrates behave like NDRG1 and TSC2, i.e. are phosphorylated by both Akt and SGK3. We have undertaken this analysis, shown in Supplementary Fig.7. The immunoblot analysis of ZR-75-1 total cell extracts undertaken with the p-Akt motif antibody suggests that 6 out of the 9 detected Akt substrates are likely to be phosphorylated by both Akt and SGK3 (SFig 7). Three Akt substrates were only phosphorylated by Akt and we observed no evidence for any SGK3 selective substrate that was not phosphorylated by Akt (SFig 7). As requested we have now modified our discussion to mention that it would be important to understand what makes a substrate a selective SGK3 substrate versus a dual SGK3/Akt substrate, and suggested that phosphorylation site consensus motifs and localisation of substrates might contribute to this.

5. The Referee requested we show the body weights for mice in Fig. 8. This data has now been included and is shown in Fig. 8d.

1. (minor) The Referee asked why inhibition of the phosphorylation of the T-loop of SGK3 would impact the phosphorylation of the hydrophobic motif (Fig. 3c). It is well known that AGC kinases are activated by dual phosphorylation of the T-loop and hydrophobic motifs and this results in a major conformational change that stabilises these enzymes and also protects the phosphorylation sites from dephosphorylation by protein phosphatases. Inhibition of the phosphorylation of either the T-loop or hydrophobic motif site is likely to promote dephosphorylation of the other site through this mechanism. Consistent with this, previous work has shown that treatment of cells with PDK1 inhibitors induces dephosphorylation of the hydrophobic motif of SGK1, S6K1, and Akt1 (Najafov A, et al. *Biochem J.* 2011;433:357-69.).

2. As requested we clarify that the phosphospecific antibody used for p-4EBP1 recognises phosphorylated Ser65. In Fig. 6 we demonstrate that following prolonged incubation of cells with the Akt inhibitor MK-2206 the phosphorylation of 4EBP1 at Ser65 increases at the same extent that mTORC1 is becoming reactivated by SGK3, explaining why it goes up. This is now emphasised better in the text.

3. As noted by Referee 1 (point 6): The Referee asked us to discuss the discrepancy between 14h ability to inhibit phosphorylation of S6K1 versus 4EBP1. We have now done this in the Discussion

and referred to several other papers in the literature where it has been shown that inhibition of mTORC1 resulting from conditions such as starvation or treatment with Rapamycin results in different degrees of dephosphorylation of various mTORC1 substrates, including 4EBP1 and S6K1. David Sabatini suggested that different mTORC substrates “encode” their sensitivity for phosphorylation by residual mTORC1 activity.

4. As noted by Referee 1 (point 3): The Referee asked why we observe that MK-2206 induces a slight inhibition of SGK3 activity (Fig. 2b). We did not comment on this in the previous version of the manuscript but we have now emphasised this observation in the Results section and stated that the underlying mechanism accounting for this observation is unclear. This is seen with both MK-2206 Akt inhibitor as well as a structurally unrelated AZD5363 Akt inhibitor suggesting that this is unlikely to be an off target effect (Fig. 2b). Further work will be required to understand the mechanism lying behind this observation. It should be noted that this effect of Akt inhibitors suppressing SGK3 activity in ZR-75-1 cells is transient and is observed after 1 hour but not after 12 hours (Fig. 6).

We would like to thank all 3 Referees for their very thoughtful comments on the manuscript, which we have tried our best to address in the 3-month period provided to revise the paper. We believe our manuscript is greatly improved as a result of the review process.

2nd Editorial Decision

14 June 2016

Thank you for submitting the revised version of your manuscript. It has now been seen by the three original referees, whose comments are enclosed below.

As you will see all referees find that their concerns have been sufficiently addressed and are broadly in favour of publication.

Given the referees' positive recommendations, I would in principle be happy to go ahead with this manuscript as soon as possible, pending some minor formal revisions and editorial issues concerning text and figures that I need you to address (please see details enclosed).

Thus I invite you to submit a revised version of the manuscript using the link provided below.

As you have probably seen already, every paper now includes a 'Synopsis', displayed on the html and freely accessible to all readers. The synopsis includes a 'model' figure - please provide a jpeg file 550 px-wide x 400-px high - as well as 2-5 one-short-sentence bullet points that summarize the article. I would appreciate if you could provide this figure and the bullet points.

Please contact me at any time if you need any help or should have further questions.

Thank you again for giving us the chance to consider your manuscript for The EMBO Journal, I look forward to the revised final version of your manuscript.

REFEREE REPORTS

Referee #1:

The authors have have addressed all my comments appropriately.

Referee #2:

The authors have satisfactorily addressed the stated concerns. This is an improved and important

study, with just one small outstanding item related to clarity below.

Despite saying that it was addressed in the response letter, the small issue brought up regarding the title was not corrected. In the title, the authors state the conclusion that the VPS34-SGK3 pathway "stimulates" tumor growth upon inhibition of the PI3K-Akt pathway. However, in the one tumor study presented (8C), Akt inhibitors slow tumor growth, so even if SGK3 is induced in these tumors, growth is not being stimulated. A reasonable and simple change would be to swap "stimulate" with "maintain", as in "to maintain mTORC1 activity and tumor growth".

Referee #3:

The authors have addressed all of my previous comments and I think the article is suitable for publication. One minor mistake, in Fig. 6, I think total 4EBP1 is mislabeled as p4EBP1.

2nd Revision - authors' response

19 June 2016

Thank you for sending us the decision letter for Manuscript EMBOJ-2016-93929R on 14th June.

Please find attached our revised manuscript in which we have altered the title as suggested by Referee 2 and corrected the Typo on Fig 6 (Referee 3).

As requested we have altered the figures and now include 4 EV figures, 1 EV Table and combined the remainder of the supplementary Figures and Tables into an Appendix that we have formatted as requested. We have also included a synopsis in power-point format with figure as Slide 1 and accompanying text as Slide 2. We have also provided Source data for Figure 2A and Fig EV4C.

We hope that the paper is now acceptable.

YOU MUST COMPLETE ALL CELLS WITH A PINK BACKGROUND ↓
PLEASE NOTE THAT THIS CHECKLIST WILL BE PUBLISHED ALONGSIDE YOUR PAPER

Corresponding Author Name: Dario R. Alessi
Journal Submitted to: EMBO Journal
Manuscript Number:EMBOJ-2016-93929R